# LaM-SLidE: Latent Space Modeling of Spatial Dynamical Systems via Linked Entities

**Florian Sestak**[1]     **Artur P. Toshev**[2]     **Andreas Fürst**[1]
**Günter Klambauer**[*,1,4]   **Andreas Mayr**[*,1]   **Johannes Brandstetter**[*,1,3]

* Equal contribution

[1] ELLIS Unit, LIT AI Lab, Institute for Machine Learning, JKU Linz, Austria
[2] Department of Engineering Physics and Computation, TUM, Germany
[3] Emmi AI GmbH, Linz, Austria,   [4] Clinical Research Institute for Medical AI, JKU Linz, Austria
{klambauer,mayr,brandstetter}@ml.jku.at

## Abstract

Generative models are spearheading recent progress in deep learning, showcasing strong promise for trajectory sampling in dynamical systems as well. However, whereas latent space modeling paradigms have transformed image and video generation, similar approaches are more difficult for most dynamical systems. Such systems – from chemical molecule structures to collective human behavior – are described by interactions of entities, making them inherently linked to connectivity patterns, entity conservation, and the traceability of entities over time. Our approach, LAM-SLIDE (Latent Space Modeling of Spatial Dynamical Systems via Linked Entities), bridges the gap between: (1) keeping the traceability of individual entities in a latent system representation, and (2) leveraging the efficiency and scalability of recent advances in image and video generation, where pre-trained encoder and decoder enable generative modeling directly in latent space. The core idea of LAM-SLIDE is the introduction of identifier representations (IDs) that enable the retrieval of entity properties and entity composition from latent system representations, thus fostering traceability. Experimentally, across different domains, we show that LAM-SLIDE performs favorably in terms of speed, accuracy, and generalizability. Code is available at `https://github.com/ml-jku/LaM-SLidE`.

## 1 Introduction

Understanding dynamical systems represents a fundamental challenge across numerous scientific and engineering domains [47, 46, 73]. In this work, we address *spatial* dynamical systems, characterized by scenes of distinguishable entities at defined spatial coordinates. Modeling temporal trajectories of such entities quickly becomes challenging, especially when stochasticity is involved. A prime example is molecular dynamics [47], where trajectories of individual atoms are modeled via Langevin dynamics, which accounts for omitted degrees of freedom by using stochastic differential equations. Consequently, the trajectories of the atoms themselves become non-deterministic.

A conventional approach to predict spatial trajectories of entities is to represent scenes as neighborhood graphs and to subsequently process these graphs with graph neural networks (GNNs). When using GNNs [83, 64, 32, 11, 86], each entity is usually represented by a node, and the spatial entities nearby are connected by an edge in the neighborhood graph. Neighborhood graphs have extensively been used for trajectory prediction tasks [51], especially for problems with a large number of indistinguishable entities, [e.g., 81, 63]. Recently, GNNs have been integrated into generative modeling frameworks to effectively capture the behavior of stochastic systems [98, 22].

39th Conference on Neural Information Processing Systems (NeurIPS 2025).

Despite their widespread use in modeling spatial trajectories, GNNs hardly follow recent trends in latent space modeling, where unified representations [45] together with universality and scalability of transformer blocks [91] offer simple application across datasets and tasks, a behavior commonly observed in computer vision and language processing [24, 25]. Notably, recent breakthroughs in image and video generation can be attributed to latent space generative modeling [38, 15]. In such paradigms, pre-trained encoders and decoders are employed to map data into a latent space, where subsequent modeling is performed, leveraging the efficiency and expressiveness of this representation. This poses the question:

> Can we leverage recent techniques from **generative latent space modeling** to boost the **modeling of stochastic trajectories of dynamical systems** with a varying number of entities?

Recently, it has been shown [5] that it is possible to model the bulk behavior of large particle systems purely in the latent space, at the cost of sacrificing the traceability of individual particles, which is acceptable or even favorable for systems where particles are indistinguishable, but challenging for, e.g., molecular dynamics, where understanding the dynamics of individual atoms is essential.

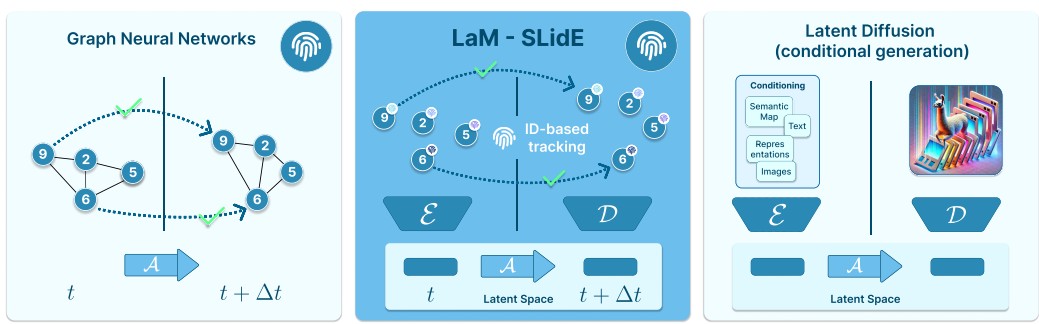

Figure 1: Overview of our approach. **Left:** Conventional graph neural networks (GNNs) model time-evolving systems (e.g., molecular dynamics) by representing entities as nodes and iteratively updating node embeddings and positions to capture system dynamics across timesteps. **Right:** Latent diffusion models employ an encoder-decoder architecture to compress input data into a lower-dimensional latent space where generative modeling is performed. Latent diffusion models, frequently enhanced with conditional information such as text, excel at generative tasks; however, due to their fixed input/output structure, they are not directly adaptable to physical systems with a varying number of entities. **Middle:** Our proposed approach LAM-SLIDE bridges these paradigms by: (1) introducing identifiers that allow traceability of individual entities, and (2) leveraging a latent system representation.

To leverage a latent system's representation, we need to be able to trace individual entities within the system. The core idea of LAM-SLIDE is the introduction of *identifiers* (IDs) that allow for the retrieval of entity properties, e.g., entity coordinates, from the latent system representation. Consequently, we can train generative models, like flow-matching [54, 57, 2], purely in latent space, where pre-trained decoder blocks map the generated representations back to the physics domain. An overview is given in Section 1. Qualitatively, LAM-SLIDE demonstrates flexibility and performs favorably across a variety of different modeling tasks.

Summarizing our contributions:

- **Latent system representation**: We propose LAM-SLIDE for generative modeling of stochastic trajectories, leveraging a latent system representation.
- **Entity structure preservation**: We introduce entity structure preservation to recover the encoded system states from the latent space representation via assignable identifiers.
- **Cross-domain generalization**: We perform experiments in different domains with varying degrees of difficulty, focusing on molecular dynamics, human motion behavior, and particle systems. LAM-SLIDE performs favorably with respect to all other architectures and showcases scalability with model size.

## 2 Background & Related Work

**Dynamical systems**. Formally, we consider a random dynamical system to be defined by a state space $\mathcal{S}$, representing all possible configurations of the system, and an evolution rule $\Phi : \mathbb{R} \times \mathcal{S} \mapsto \mathcal{S}$ that determines how a state $\mathbf{s} \in \mathcal{S}$ evolves over time, and which exhibits the following properties for the time differences $0, \hat{t}_1$, and, $\hat{t}_2$:

$$\Phi(0, \mathbf{s}) = \mathbf{s} \tag{1}$$

$$\Phi(\hat{t}_2, \Phi(\hat{t}_1, \mathbf{s})) = \Phi(\hat{t}_1 + \hat{t}_2, \mathbf{s}) \tag{2}$$

We note that $\Phi$ does not necessarily need to be defined on the whole space $\mathbb{R} \times \mathcal{S}$, but we assume this for notational simplicity. The exact formal definition of random dynamical systems is more involved and consists of a base flow (noise) and a cocycle dynamical system defined on a physical phase space [8]. We skip the details, but assume that we deal with random dynamical systems for the remainder of the paper. The non-deterministic behavior of such dynamical systems suggests the use of generative modeling approaches.

**Generative modeling**. Recent developments in generative modeling have captured widespread interest. The breakthroughs of the last years were mainly driven by diffusion models [87, 88, 37], a paradigm that transforms a simple distribution into a target data distribution via iterative refinement steps. Flow Matching [54, 57, 2] has emerged as a powerful alternative to diffusion models, enabling simulation-free training between arbitrary start and target distributions [55]. It has also been extended to data manifolds [19]. This approach comes with straighter paths, offering faster integration, and has been successfully applied across different domains like images [27], audio [92], videos [71], protein design [41], and robotics [13].

**Latent space modeling**. Latent space modeling has achieved remarkable success at image and video generation [15, 27], where pre-trained encoders and decoders map data into a latent space, and back into the physics space. The latent space aims to preserve the essential structure and features of the original data, often following a compositional structure $\mathcal{D} \circ \mathcal{A} \circ \mathcal{E}$ [85, 4, 5], where the encoder $\mathcal{E}$ maps the input signal into the latent space, the approximator $\mathcal{A}$ models a process, and the decoder maps back to the original space. Examples of approximators include conditional generative modeling techniques, such as generating an image given a text prompt (condition) [77]. This framework was recently used for 3D shape generation, where the final shape in the spatial domain is then constructed by querying the latent representations over a fixed spatial grid [101, 100].

## 3 LaM - SLidE

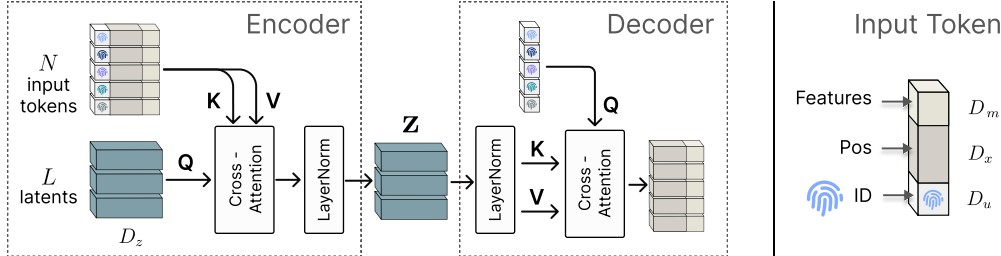

Figure 2: Architecture of our encoder-decoder structure (First Stage): **Left:** The encoder maps $N$ input tokens to a latent system representation by cross-attending to $L$ learned latent query tokens. The decoder reconstructs the input data from the latent representation using the assigned IDs. **Right:** Structure of the input token, consisting of an ID, spatial information, and features (see also Figure 3).

We introduce an *identifier pool* and an *identifier assignment function* which allows us to effectively map and retrieve entities to and from a latent system representation. The ID components preserve the relationships between entities, making them traceable across time-steps. LaM-SLidE follows an encoder $\mathcal{E}$ - approximator $\mathcal{A}$ - decoder $\mathcal{D}$ paradigm.

### 3.1 Problem Formulation

**State space**. We consider spatial dynamics. Our states $\mathbf{s} \in \mathcal{S}$ describe the configuration of entities within the scene together with their individual features. We assume that a scene consists of $N$ entities $e_i$ with $i \in 1, \ldots, N$. An entity $e_i$ is described by its spatial location $\mathbf{x}_i \in \mathbb{R}^{D_x}$ and some further properties $\mathbf{m}_i \in \mathbb{R}^{D_m}$ (e.g., atom type, etc.). We denote the set of entities as $E = \{e_1, \ldots, e_n\}$. We consider states $\mathbf{s}^t$ at discretized timepoints $t$. Analogously, we use $\mathbf{x}_i^t$, $\mathbf{m}_i^t$ to describe coordinates and properties at time $t$, respectively. We refer to the coordinate concatenation $[\mathbf{x}_1^t, .., \mathbf{x}_N^t]$ of the $N$ entities in $\mathbf{s}^t$ as $\mathbf{X}^t \in \mathbb{R}^{N \times D_x}$. Analogously, we use $\mathbf{M}^t \in \mathbb{R}^{N \times D_m}$ to denote $[\mathbf{m}_1^t, .., \mathbf{m}_N^t]$. When properties are conserved over time, i.e., $\mathbf{M}^t = \mathbf{M}^1$, we just skip the time index and the time-wise repetition of states and use $\mathbf{M} \in \mathbb{R}^{N \times D_m}$. We concatenate sequences of coordinate states $\mathbf{X}^t$ with $t \in 1 .. T$ to a tensor $\mathbf{X} \in \mathbb{R}^{T \times N \times D_x}$, which describes a whole sampled coordinate trajectory of a system with $T$ time points and $N$ entities. An example of such trajectories from dynamical systems are molecular dynamics trajectories (e.g. Figure 15). Notation is summarized in Table 6.

**Prediction task**. We predict a trajectory of entity coordinates $\mathbf{X}^{[T_o+1:\, T]} = [\mathbf{X}^{T_o+1}, \ldots, \mathbf{X}^t, \ldots, \mathbf{X}^T] \in \mathbb{R}^{(T-T_o) \times N \times D_x}$, given a short (observed) initial trajectory $\mathbf{X}^{[1:\, T_o]} = [\mathbf{X}^1, \ldots, \mathbf{X}^t, \ldots, \mathbf{X}^{T_o}] \in \mathbb{R}^{T_o \times N \times D_x}$ together with general (time-invariant) entity properties $\mathbf{M}$. Here, $T_o$ denotes the length of the observed trajectory and $T_f = T - T_o$ denotes the prediction horizon.

### 3.2 Entity Structure Preservation

We aim to preserve the integrity of individual entities in a latent system representation. More specifically, we aim to preserve both the number of entities as well as their structure. For example, in the case of molecules, we want to preserve both the number of atoms and the atom composition. We therefore assign identifiers from an identifier pool to each entity, allowing us to trace the entities by their assigned identifiers. The two key components are: (i) creating a fixed, finite pool of *identifiers (IDs)* and (ii) defining a unique mapping between entities and identifiers.

**Definition 3.1.** For a fixed $u \in \mathbb{N}$, let $\mathcal{I} = \{0, 1, \ldots, u-1\}$ be the *identifier pool*. An *identifier* $i$ is an element of the set $\mathcal{I}$.

**Definition 3.2.** Let $E$ be a finite set of entities and $\mathcal{I}$ an identifier pool. The *identifier assignment pool* is the set of all injective functions from $E$ to $\mathcal{I}$:

$$I = \{\texttt{ida}(\cdot) : E \mapsto \mathcal{I} \mid \forall e_i, e_j \in E : e_i \neq e_j \implies \texttt{ida}(e_i) \neq \texttt{ida}(e_j)\}, \tag{3}$$

An *identifier assignment function* $\texttt{ida}(\cdot)$ is an element of the set $I$.

**Proposition 3.3.** *Given an identifier pool $\mathcal{I}$ and a finite set of entities $E$, an identifier assignment pool $I$ as defined by Definition 3.2 is non-empty if and only if $|E| \leqslant |\mathcal{I}|$.*

Proof, see App. F.1.

**Proposition 3.4.** *Given an identifier pool $\mathcal{I}$ and a finite set of entities $E$ such that $|E| \leqslant |\mathcal{I}|$, the identifier assignment pool $I$ as defined by Definition 3.2 contains finitely many injective functions.*

Proof, see App. F.2.

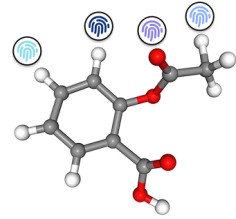

Notably, since $\texttt{ida}(\cdot)$ may be selected randomly – the specific choice of the mapping $\texttt{ida}(\cdot)$ can be arbitrary – the only requirement is that an injective mapping between entities and identifiers is established, i.e., each entity is uniquely assigned to an identifier, but not all identifiers need to be assigned to an entity. Further, Proposition 3.3 suggests using an identifier pool that is large enough, such that a model learned on this identifier pool can generalize across systems with varying numbers of entities.

**Example aspirin**. Aspirin $C_9H_8O_4$ consists of 21 atoms, thus the identifier pool $\mathcal{I}$ needs to have at least 21 unique identifiers. We select an assignment function $\texttt{ida}(\cdot)$, arbitrary, and use it to assign each atom a unique identifier. Notably, e.g., for molecules, we do not explicitly model molecular bond information, as the spatial relationship between atoms (interatomic distances) implicitly captures this information. Figure 3 shows an arbitrary but fixed identifier assignment for aspirin; we illustrate different IDs by colored fingerprint symbols.

Figure 3: **Example aspirin**: IDs are assigned to the atoms of the molecule.

## 3.3 Model Architecture

Since predicting continuations of system trajectories is a conceptually similar task to generating videos from an initial sequence of images, we took inspiration from Blattmann et al. [15] in using a latent diffusion architecture. We also took inspiration from Jaegle et al. [43] to decompose our model architecture as follows: To map the state of the system composed of $N$ entities to a latent space containing $L$ learned latent tokens ($\in \mathbb{R}^{D_z}$), we use a cross-attention mechanism. In the resulting latent space, we aim to train an approximator to predict future latent states based on the embedded initial states. Inversely to the encoder, we again use a cross-attention mechanism to retrieve the physical information of the individual entities of the system from the latent system representation. To wrap it up, LAM-SLIDE , is built up by an encoder $\mathcal{E}$ - approximator $\mathcal{A}$ - decoder $\mathcal{D}$ architecture, $\mathcal{D} \circ \mathcal{A} \circ \mathcal{E}$. A detailed composition of $\mathcal{E}$ and $\mathcal{D}$ is shown in Figure 2.

**Encoder**. The encoder $\mathcal{E}$ aims to encode a state of the system such that the properties of each entity $e_n$ can be decoded (retrieved) later. At the same time, the structure of the latent state representation $\mathbf{Z}^t \in \mathbb{R}^{L \times D_z}$ is constant and should not depend on the different number $N$ of entities. This contrasts with GNNs, where the number of latent vectors depends on the number of nodes.

To allow for traceability of the entities, we first embed each identifier $i$ in the space $\mathbb{R}^{D_u}$ by a learned embedding $\texttt{IDEmb} : \mathcal{I} \mapsto \mathbb{R}^{D_u}$. We map all $(n = 1, \ldots, N)$ system entities $e_n$ to $\mathbf{u}_n \in \mathbb{R}^{D_u}$ as follows:

$$\texttt{ida}(\cdot) \qquad \text{(arbitrary identifier assignment)} \qquad (4)$$

$$\mathbf{u}_n = \texttt{IDEmb}(\texttt{ida}(e_n)) \qquad \forall n \in 1, \ldots, N \qquad (5)$$

The inputs to the encoder comprise the time dependent location $\mathbf{x}_n^t \in \mathbb{R}^{D_x}$, properties $\mathbf{m}_n \in \mathbb{R}^{D_x}$, and identity representation $\mathbf{u}_n \in \mathbb{R}^{D_x}$ of each entity $e_n$, as visualized in the right part of Figure 2. We concatenate the different types of features across all entities of the system: $\mathbf{X}^t = [\mathbf{x}_1^t, .., \mathbf{x}_N^t]$, $\mathbf{M} = [\mathbf{m}_1, .., \mathbf{m}_N]$ and $\mathbf{U} = [\mathbf{u}_1, .., \mathbf{u}_N]$.

The encoder $\mathcal{E}$ maps the input to a latent system representation via

$$\mathcal{E} : [\mathbf{X}^t, \mathbf{M}, \mathbf{U}] \in \mathbb{R}^{N \times (D_x + D_m + D_u)} \mapsto \mathbf{Z}^t \in \mathbb{R}^{L \times D_z},$$

realized by cross-attention [91, 75] between the input tensor $\in \mathbb{R}^{N \times (D_x + D_m + D_u)}$, which serves as keys and values, and a fixed number of $L$ learned latent vectors $\in \mathbb{R}^{D_z}$ [43], which serve as the queries. The encoding process is depicted on the left side of Figure 2.

**Decoder**. The aim of the decoder $\mathcal{D}$ is to retrieve the system state information $\mathbf{X}^t$ and $\mathbf{M}$ from the latent state representation $\mathbf{Z}^t$ using the encoded entity identifier embeddings $\mathbf{U}$. The decoder $\mathcal{D}$ maps the latent system representation back into the coordinates and properties of each entity via

$$\mathcal{D} : \mathbf{Z}^t \in \mathbb{R}^{L \times D_z} \times \mathbf{U} \in \mathbb{R}^{L \times D_u} \mapsto [\mathbf{X}^t, \mathbf{M}] \in \mathbb{R}^{N \times (D_x + D_m)}.$$

As shown in the middle part of Figure 2, $\mathcal{D}$ is realized by cross-attention layers. The latent space representation $\mathbf{Z}^t$ serves as the keys and values in the cross-attention mechanism, while the embedded identifier $\mathbf{u}_n$ acts as the query. Applied to to all $(n = 1, \ldots, N)$ system entities $e_n$, this results in the retrieved system state information $\mathbf{X}^t$ and $\mathbf{M}$. Using the learned identifier embeddings as queries can be interpreted as a form of content-based retrieval and associative memory [6, 40, 75].

**Approximator**. Finally, the approximator models the system's time evolution in latent space, i.e., it predicts a series of future latent system states $\mathbf{Z}^{[T_o+1: T]} = [\mathbf{Z}^{T_o+1}, \ldots, \mathbf{Z}^t, \ldots, \mathbf{Z}^T]$, given a series of initial latent system states $\mathbf{Z}^{[1: T_o]} = [\mathbf{Z}^1, \ldots, \mathbf{Z}^t, \ldots, \mathbf{Z}^{T_o}]$,

$$\mathcal{A} : \mathbf{Z}^{[1: T_o]} \in \mathbb{R}^{T_o \times L \times D_z} \mapsto \mathbf{Z}^{[T_o+1: T]} \in \mathbb{R}^{T_f \times L \times D_z}.$$

Given the analogy of predicting the time evolution of a dynamical system to the task of synthesizing videos, we realized $\mathcal{A}$ by a flow-based model. Specifically, we constructed it based on the stochastic interpolants framework [2, 60].

We are interested in time-dependent processes, which interpolate between data $\mathbf{o}_1 \sim p_1$ from a target data distribution $p_1$ and noise $\epsilon \sim p_0 := \mathcal{N}(\mathbf{0}, \boldsymbol{I})$:

$$\mathbf{o}_\tau = \alpha_\tau \mathbf{o}_1 + \sigma_\tau \epsilon, \qquad (6)$$

where $\tau \in [0, 1]$ is the time parameter of the flow (to be distinguished from system times $t$). $\alpha_\tau$ and $\sigma_\tau$ are differentiable functions in $\tau$, which have to fulfill $\alpha_\tau^2 + \sigma_\tau^2 > 0 \ \forall \tau \in [0, 1]$, and, further $\alpha_0 = \sigma_1 = 0$, and, $\alpha_1 = \sigma_0 = 1$. The goal is to learn a parametric model $v_\theta(\mathbf{o}, \tau)$, s.t., $\int_0^1 \mathbb{E}[||v_\theta(\mathbf{o}_\tau, \tau) - \dot{\alpha}_\tau \mathbf{o}_1 - \dot{\sigma}_\tau \epsilon||^2] \, d\tau$ is minimized. Within the stochastic interpolants framework, we identify $\mathbf{o}_1$ with a whole trajectory $\mathbf{Z} = \mathbf{Z}^{[1:\,T]} = [\mathbf{Z}^{[1:\,T_o]}, \mathbf{Z}^{[T_o+1:\,T]}] \in \mathbb{R}^{T \times L \times D_z}$.

Since the generated trajectories should be conditioned on the latent system representations of initial time frames $\mathbf{Z}^{[1:\,T_o]}$, we extend $v_\theta$ with a conditioning argument $\mathbf{C} \in \mathbb{R}^{T \times L \times D_z}$, making it effectively a conditional vector field $v_\theta : \mathbb{R}^{T \times L \times D_z} \times [0, 1] \times \mathbb{R}^{T \times L \times D_z} \mapsto \mathbb{R}^{T \times L \times D_z}$. The tensor structure of $\mathbf{C}$ is the same as the one for $\mathbf{Z}$. For the first time steps, both tensors have equal values, i.e., $\mathbf{C}^{[1:\,T_o]} = \mathbf{Z}^{[1:\,T_o]}$. The remaining tensor entries $\mathbf{C}^{[T_o+1:\,T]}$ are filled up with mask tokens, see Figure 4. The latent model is structured as series transformer blocks [90, 67], alternating between the spatial and temporal dimensions (see App. B.4). We parametrized our model using a data prediction objective [55], see App. C.1. For architectural details see App. B.

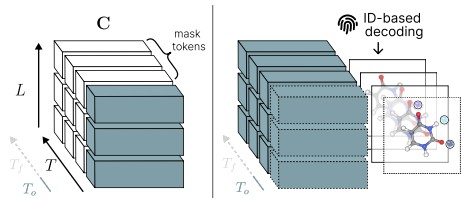

Figure 4: **Left:** The latent model receives conditioning via known tokens (observed timesteps) and mask tokens (for prediction). This example shows conditioning on one time-frame to predict three future ones. **Right:** ID-based decoding, where the predicted atom positions are decoded by the assigned IDs.

### 3.4 Training Procedure

The training process follows a two-stage approach similar to latent diffusion models [77]. **First Stage:** We train the encoder $\mathcal{E}$ and decoder $\mathcal{D}$, to reconstruct entities from latent space using assigned IDs, see Figure 2). **Second Stage:** We train the approximator $\mathcal{A}$ on the latent system representations produced by the frozen encoder $\mathcal{E}$ (details in App. E.3 and Figure 5).

## 4 Experiments

In order to evaluate LAM-SLIDE we focus on three key aspects: (i) **Robust generalization in diverse domains**. We examine LAM-SLIDE 's generalization in different data domains in relation to other methods, for which we utilize tracking data from human motion behavior, trajectories of particle systems, and data from *molecular dynamics* (MD) simulations. (ii) **Temporal adaptability**. We evaluate temporal adaptability through various conditioning/prediction horizons, considering single/multi-frame conditioning and short/long-term predictions; (iii) **Computational efficiency and scalability**. Finally, we assess the inference time of LAM-SLIDE , and evaluate performance with respect to model size. The subsequent sections show our key findings. For implementation details and additional results see App. E and G, for information on the datasets see App. E.1.

**Metrics**. We utilized the Average Discrepancy Error (ADE) and the Final Discrepancy Error (FDE), defined as $\mathrm{ADE}(\mathbf{X}, \hat{\mathbf{X}}) = \frac{1}{(T-T_o)N} \sum_{t=T_o+1}^{T} \sum_{i=1}^{N} \|\mathbf{X}_i^t - \hat{\mathbf{X}}_i^t\|_2$, $\mathrm{FDE}(\mathbf{X}, \hat{\mathbf{X}}) = \frac{1}{N} \sum_{i=1}^{N} \|\mathbf{X}_i^T - \hat{\mathbf{X}}_i^T\|_2$, capturing model performance across predicted future time steps and the model performance specifically for the last predicted frame, respectively. These metrics represent well-established evaluation criteria in trajectory forecasting [94, 95].

For the MD experiments on peptides (Tetrapeptides), we use the Jensen-Shannon divergence (JSD) over the distribution of torsion angles, considering both backbone (BB) and side chain (SC) angles. To capture long temporal behavior, we use *Time-lagged Independent Component Analysis* (TICA) [70], focusing on the slowest components TIC 0 and TIC 1. To investigate metastable state transitions, we make use of *Markov State Models* (MSMs) [74, 65].

For inference time and scalability, we assess the *number of function evaluations* (NFE) and report the performance of our method for different model sizes. A comprehensive overview of the conditioning and prediction horizons for each experiment is shown in Table 7.

Table 1: **Results on generation on N-body dataset**, in terms of ADF/FDE averaged over 5 runs.

|  | Particle | | Spring | | Gravity | |
|---|---|---|---|---|---|---|
|  | ADE | FDE | ADE | FDE | ADE | FDE |
| RF [52][a] | 0.479 | 1.050 | 0.0145 | 0.0389 | 0.791 | 1.630 |
| TFN [89][a] | 0.330 | 0.754 | 0.1013 | 0.2364 | 0.327 | 0.761 |
| SE(3)-Tr [30][a] | 0.395 | 0.936 | 0.0865 | 0.2043 | 0.338 | 0.830 |
| EGNN [82][a] | 0.186 | 0.426 | 0.0101 | 0.0231 | 0.310 | 0.709 |
| EqMotion [94][a] | 0.141 | 0.310 | 0.0134 | 0.0358 | 0.302 | 0.671 |
| SVAE [95][a] | 0.378 | 0.732 | 0.0120 | 0.0209 | 0.582 | 1.101 |
| GeoTDM [35][a] | 0.110 | 0.258 | **0.0030** | **0.0079** | 0.256 | 0.613 |
| LAM-SLIDE | **0.104** | **0.238** | 0.0070 | 0.0135 | **0.157** | **0.406** |

[a] Results from Han et al. [35].

Table 2: **Results on the ETH-UCY** dataset for pedestrian forecasting. Results are in terms of minADE/minFDE out of 20 runs.

|  | ETH | Hotel | Univ | Zara1 | Zara2 | Average |
|---|---|---|---|---|---|---|
| Linear[a] | 1.07/2.28 | 0.31/0.61 | 0.52/1.16 | 0.42/0.95 | 0.32/0.72 | 0.53/1.14 |
| SGAN [34][a] | 0.64/1.09 | 0.46/0.98 | 0.56/1.18 | 0.33/0.67 | 0.31/0.64 | 0.46/0.91 |
| SoPhie [78][a] | 0.70/1.43 | 0.76/1.67 | 0.54/1.24 | 0.30/0.63 | 0.38/0.78 | 0.54/1.15 |
| PECNet [61][a] | 0.54/0.87 | 0.18/0.24 | 0.35/0.60 | 0.22/0.39 | 0.17/0.30 | 0.29/0.48 |
| Traj++ [80][a] | 0.54/0.94 | 0.16/0.28 | 0.28/0.55 | 0.21/0.42 | 0.16/0.32 | 0.27/0.50 |
| BiTraP [97][a] | 0.56/0.98 | 0.17/0.28 | 0.25/0.47 | 0.23/0.45 | 0.16/0.33 | 0.27/0.50 |
| MID [33][a] | 0.50/0.76 | 0.16/0.24 | 0.28/0.49 | 0.25/0.41 | 0.19/0.35 | 0.27/0.45 |
| SVAE [95][a] | 0.47/0.76 | 0.14/0.22 | 0.25/0.47 | **0.20**/0.37 | **0.14/0.28** | **0.24**/0.42 |
| GeoTDM [35][a] | 0.46/0.64 | **0.13**/0.21 | **0.24/0.45** | 0.21/0.39 | 0.16/0.30 | **0.24/0.40** |
| LAM-SLIDE | **0.45**/0.75 | **0.13/0.19** | 0.26/0.47 | 0.21/0.35 | 0.17 / 0.30 | **0.24**/ 0.41 |

[a] Results from Han et al. [35].

## 4.1 Pedestrian Trajectory Forecasting (ETH-UCY)

**Experimental setup**. For human motion behavior, we first consider the ETH-UCY dataset [68, 53], which provides pedestrian movement behavior, over five different scenes: ETH, Hotel, Univ, Zara1, and Zara2. We use the same setup as Han et al. [35], Xu et al. [94, 95], in which the methods obtain the first 8 frames as conditioning information and have to predict the next 12 frames. We report the minADE/minFDE, computed across 20 sampled trajectories.

**Compared methods**. We compare LAM-SLIDE to eight state-of-the-art generative methods covering different model categories, including: GANs : SGAN [34], SoPhie [78]; VAEs: PECNet [61], Traj+ + [80], BiTrap [97], SVAE [95]; diffusion models: MID [33] and GeoTDM [35] and a linear baseline. The baseline methods predominantly target pedestrian trajectory prediction, with GeoTDM and Linear being the exceptions.

**Results**. As shown in Table 2, our model performs competitively across all five scenes, achieving lower minFDE for Zara1 and Hotel scene, and the lowest minADE on the ETH scene. Notably, in contrast to the compared baselines, we did not create additional features like velocity and acceleration or imply any kind of connectivity between entities.

## 4.2 Basketball Player Trajectory Forecasting (NBA)

**Experimental setup**. Our second experiment examines human motion behavior in the context of basketball gameplay. We utilize the SportVU NBA movement dataset [99], which contains player movement data from the 2015-2016 NBA season. Each recorded frame includes ten player positions (5 for each team) and the ball position. We consider two different scenarios: a) *Rebounding:* containing scenes of missed shots; b) *Scoring:* containing scenes of a team scoring a basket. The interaction patterns – both in frequency and in their adversarial versus cooperative dynamics – exhibit different challenges as those in Section 4.1. The evaluation procedure by Xu et al. [95], which we use, provides 8 frames as input conditioning and the consecutive 12 frames for prediction. The performance is reported by the minADE/minFDE metrics, which are computed across 20 sampled trajectories. For the reported metrics, only the trajectories of the players are considered.

Table 3: **Results on the NBA dataset:** Compared methods have to predict player positions for 12 frames and are given the initial 8 frames as input. Results in terms of minADE/minFDE for the Rebounding and Scoring scenarios.

|  | Rebounding | Scoring |
|---|---|---|
| Linear[a] | 2.14/5.09 | 2.07/4.81 |
| Traj++ [80][a] | 0.98/1.93 | 0.73/1.46 |
| BiTraP [97][a] | 0.83/1.72 | 0.74/1.49 |
| SGNet-ED [93][a] | 0.78/1.55 | 0.68/1.30 |
| SVAE [95][a] | **0.72/1.37** | **0.64**/1.17 |
| LAM-SLIDE | 0.79/1.42 | **0.64/1.09** |

[a] Results from Xu et al. [95].

**Compared methods**. The method comparison for this human motion forecasting task includes methods based on VAEs, Traj++ [80], BiTrap [97], SGNet-ED [93], SVAE [95], and a linear baseline.

**Results**. As illustrated in Table 3, our model shows robust performance across both scenarios, Rebounding and Scoring. For the Scoring scenario, LAM-SLIDE achieves parity with SocialVAE [34]

in terms of minADE and surpasses the performance in terms of minFDE. In the Rebounding scenario, we observe comparable but slightly lower performance of LAM-SLIDE compared to SocialVAE. Trajectories sampled by our model are shown in Figure 13.

## 4.3 N-Body System Dynamics (Particle Systems)

**Experimental setup**. We evaluate our method across three distinct N-Body simulation scenarios: a) *Charged Particles*: comprising particles with randomly assigned charges $+1/-1$ interacting via Coulomb forces [51, 82]; b) *Spring Dynamics*: consisting of $N = 5$ particles with randomized masses connected by springs with a probability $0.5$ between particle pairs [51]; and c) *Gravitational Systems*: containing $N = 10$ particles with randomized masses and initial velocities governed by gravitational interactions [17]. For all three scenarios, we consider 10 conditioning frames and 20 prediction frames. In line with Han et al. [35], we use 3000 trajectories for training and 2000 trajectories for validation and testing, and we report ADE/FDE averaged over $K = 5$ runs.

**Compared methods**. We compare LAM-SLIDE to seven different methods, including six equivariant GNN based methods: Tensor Field Network [89], Radial Field [52], SE(3)-Transformer [30], EGNN [82], EqMotion [94], GeoTDM [35], and a non-equivariant method: SVAE [95].

**Results**. LAM-SLIDE achieves the best performance in terms of ADE/FDE for the Charged Particles and Gravity scenarios and competitive second-rank performance in the Spring Dynamics scenario, see Table 1. Unlike compared methods, LAM-SLIDE achieves these results without computing intermediate physical quantities such as velocities or accelerations. We present sampled trajectories in Figure 14.

## 4.4 Molecular Dynamics - Small Molecules (MD17)

**Experimental setup**. In this experiment, we evaluate LAM-SLIDE on the well-established MD17 [21] dataset, containing simulated molecular dynamics trajectories of 8 small molecules. The size of those molecules ranges from 9 atoms (Ethanol and Malonaldehyde) to 21 atoms (Aspirin). We use 10 frames as conditioning, 20 frames for prediction, and report ADE/FDE averaged over $K = 5$ runs.

**Compared methods**. Similar to Section 4.3, we compared LAM-SLIDE to: Tensor Field Network [89], Radial Field [52], SE(3)-Transformer [30], EGNN [82], EqMotion [94], GeoTDM [35], and SVAE [95].

**Results**. The results in Table 4 show the performance on the MD17 benchmark. LAM-SLIDE achieves the lowest ADE/FDE of all methods and for all molecules. These results are particularly remarkable considering that: (1) our model operates without an explicit definition of molecular bond information, and (2) it surpasses the performance of all equivariant baselines, an inductive bias we intentionally omitted in LAM-SLIDE .

Notably, we train a single model on all molecules – a feat that is structurally encouraged by the design of LAM-SLIDE . For ablation, we also train GeoTDM [35] on all molecules and evaluate the performance on each one of them ("all→each" in the Table 13). Interestingly, we also observe consistent improvements in the GeoTDM performance. However, GeoTDM's performance does not reach that of LAM-SLIDE . We also note that our latent model is trained for 2000 epochs, while GeoTDM was trained for 5000 epochs. Trajectories are shown in Figure 15.

## 4.5 Molecular Dynamics - Tetrapeptides (4AA)

**Experimental setup**. To investigate LAM-SLIDE on long prediction horizons, we utilize the Tetrapeptide dataset from Jing et al. [44], containing explicit-solvent molecular dynamics trajectories simulated using OpenMM [26]. The dataset comprises 3,109 training, 100 validation, and 100 test peptides. We use a single conditioning frame to predict 10,000 consecutive frames. The predictions are structured as a sequence of ten cascading 1,000-step rollouts, where each subsequent rollout is conditioned on the final frame of the previous. Note that, in contrast to the MD17 dataset, the methods predict trajectories of *unseen* molecules.

Table 4: **Results on the MD17 dataset**. Compared methods have to predict the atom positions of 20 frames, conditioned on 10 input frames. Results in terms of ADE/FDE, averaged over 5 runs.

| | Aspirin | | Benzene | | Ethanol | | Malonaldehyde | | Naphthalene | | Salicylic | | Toluene | | Uracil | |
|---|---|---|---|---|---|---|---|---|---|---|---|---|---|---|---|---|
| | ADE | FDE | ADE | FDE | ADE | FDE | ADE | FDE | ADE | FDE | ADE | FDE | ADE | FDE | ADE | FDE |
| RF [52][a] | 0.303 | 0.442 | 0.120 | 0.194 | 0.374 | 0.515 | 0.297 | 0.454 | 0.168 | 0.185 | 0.261 | 0.343 | 0.199 | 0.249 | 0.239 | 0.272 |
| TFN [89][a] | 0.133 | 0.268 | 0.024 | 0.049 | 0.201 | 0.414 | 0.184 | 0.386 | 0.072 | 0.098 | 0.115 | 0.223 | 0.090 | 0.150 | 0.090 | 0.159 |
| SE(3)-Tr. [30][a] | 0.294 | 0.556 | 0.027 | 0.056 | 0.188 | 0.359 | 0.214 | 0.456 | 0.069 | 0.103 | 0.189 | 0.312 | 0.108 | 0.184 | 0.107 | 0.196 |
| EGNN [82][a] | 0.267 | 0.564 | 0.024 | 0.042 | 0.268 | 0.401 | 0.393 | 0.958 | 0.095 | 0.133 | 0.159 | 0.348 | 0.207 | 0.294 | 0.154 | 0.282 |
| EqMotion [94][a] | 0.185 | 0.246 | 0.029 | 0.043 | 0.152 | 0.247 | 0.155 | 0.249 | 0.073 | 0.092 | 0.110 | 0.151 | 0.097 | 0.129 | 0.088 | 0.116 |
| SVAE [95][a] | 0.301 | 0.428 | 0.114 | 0.133 | 0.387 | 0.505 | 0.287 | 0.430 | 0.124 | 0.135 | 0.122 | 0.142 | 0.145 | 0.171 | 0.145 | 0.156 |
| GeoTDM [35] [a] | _0.107_ | _0.193_ | _0.023_ | _0.039_ | _0.115_ | _0.209_ | _0.107_ | _0.176_ | _0.064_ | _0.087_ | _0.083_ | _0.120_ | _0.083_ | _0.121_ | _0.074_ | _0.099_ |
| LᴀM-SLɪᴅE | **0.059** | **0.098** | **0.021** | **0.032** | **0.087** | **0.167** | **0.073** | **0.124** | **0.037** | **0.058** | **0.047** | **0.074** | **0.045** | **0.075** | **0.050** | **0.074** |

[a] Results from Han et al. [35].

**Compared methods**.   We compare LᴀM-SLɪᴅE to the recently proposed method MDGen [44], which is geared towards protein MD simulations, and to a replicate of the ground truth MD simulation as a baseline, which is a lower bound for the performance.

**Results**.   Table 5 shows performance metrics of the methods (for details on those metrics see App. E.6). Figure 16 shows the distribution of backbone torsion angles, and the free energy surfaces of the first two TICA components, for ground truth vs simulated trajectories. LᴀM-SLɪᴅE performs competitively with the current state-of-the-art method MDGen with respect to torsion angles, which is a notable achievement given that MDGen operates in torsion space only. With respect to the TICA and MSM metrics, LᴀM-SLɪᴅE even outperforms MDGen. Sampled trajectories are shown in Figure 17.

## 4.6   Computational Efficiency and Scaling Behavior

To assess computational efficiency, we compare the NFEs of our model to the second-best method, GeoTDM. Our results show that LᴀM-SLɪᴅE requires up to 10x-100x fewer function evaluations depending on the domain. For a detailed discussion, see App. G.2.

We further assess the scalability of LᴀM-SLɪᴅE across different model sizes. Our results indicate that performance consistently improves with increasing parameter count, suggesting that our method benefits from larger model capacity and could potentially achieve even better results with additional computational resources. Comprehensive details of this analysis can be found in App. G.

Table 5: **Results on the Tetrapeptide dataset:** Columns denote the JSD between distributions of *torsion angles* (backbone (BB), side-chain (SC), and all angles), the TICA, and the MSM metric.

| | Torsions | | | TICA | | MSM | Time |
|---|---|---|---|---|---|---|---|
| | BB | SC | All | 0 | 0,1 joint | | |
| 100ns [a] | .103 | .055 | .076 | .201 | .268 | .208 | ∼ 3h |
| MDGen[44][a] | .130 | **.093** | **.109** | .230 | .316 | .235 | ∼ 60s |
| LᴀM-SLɪᴅE | **.128** | .122 | .125 | **.227** | **.315** | **.224** | ∼ 53s |

[a] Results from Jing et al. [44].

## 4.7   Ablations

**Identifier Pool Size**.   To understand how the size of the identifier pool affects the performance of the first stage model, we conducted additional experiments on the MD17 dataset. Maintaining a fixed number of update steps across different configurations results in a modest increase in the reconstruction error for larger pool sizes. We attribute this to the reduced number of updates per individual identifier embedding. A comprehensive analysis is provided in App. G.4.

**Identifier Assignment**.   We assessed the sensitivity of our model to the specific entity-identifier assignment by evaluating five random assignments on the MD17 dataset. Results do not indicate a negative impact on the performance of our model. This demonstrates the robustness of our approach with respect to the specific assignment between entities and identifiers, see App. G.5 for more details.

**Identifier Latent Utilization**.   We analyzed decoder attention patterns to understand how identifiers access latent information across different compression ratios. Our findings show that each identifier maintains a consistent addressing scheme across different molecule conformations, and the model adaptively utilizes attention heads based on the capacity of the latent space. In an overparameterized

setting, the model favors single-head retrieval, while in the compressed setting utilizes both heads to avoid potential identifier collisions. See Appendix G.6 for detailed analysis.

## 5 Discussion

**Limitations**. Our experiments indicate that our architecture applies to a diverse set of problems; however, a few limitations provide opportunities for future improvement. While our current approach successfully allows for compressing entities with beneficial reconstruction performance, our experiments indicate a tradeoff between the number of latent space vectors to encode system states and the performance of our latent model; for more details, see App. G.3.

**Conclusion**. We have introduced LAM-SLIDE, a novel approach for modeling spatial dynamical systems that consist of a variable number of entities within a fixed-size latent system representation, where assignable identifiers enable the traceability of individual entities. Across diverse domains, LAM-SLIDE matches or exceeds specialized methods and offers promising scalability properties. Its minimal reliance on prior knowledge makes it suitable for many tasks, suggesting its potential as a foundational architecture for dynamical systems.

## Acknowledgments

We thank Niklas Schmidinger, Anna Zimmel, Benedikt Alkin, and Arturs Berzins for helpful discussions and feedback. The ELLIS Unit Linz, the LIT AI Lab, and the Institute for Machine Learning are supported by the Federal State Upper Austria. We thank the projects FWF AIRI FG 9-N (10.55776/FG9), AI4GreenHeatingGrids (FFG- 899943), Stars4Waters (HORIZON-CL6-2021-CLIMATE-01-01), FWF Bilateral Artificial Intelligence (10.55776/COE12). We thank NXAI GmbH, Audi AG, Silicon Austria Labs (SAL), Merck Healthcare KGaA, GLS (Univ. Waterloo), TÜV Holding GmbH, Software Competence Center Hagenberg GmbH, dSPACE GmbH, TRUMPF SE + Co. KG.

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

# Appendix

## Table of Contents

# A  Notation

Table 6: Overview of used symbols and notations.

| Definition | Symbol/Notation | Type |
|---|---|---|
| continuous time | $\hat{t}$ | $\mathbb{R}$ |
| overall number of (sampled) time steps | $T$ | $\mathbb{N}$ |
| number of observed time steps (when predicting later ones) | $T_o$ | $\mathbb{N}$ |
| number of future time steps (prediction horizon) | $T_f = T - T_o$ | $\mathbb{N}$ |
| time index for sequences of time steps | $t$ | $\mathbb{N}$ |
| system state space | $\mathcal{S}$ | application-dependent set, to be further defined |
| system state | $\mathbf{s}$ | $\mathcal{S}$ |
| entity | $e$ | symbolic |
| number of entities | $N$ | $\mathbb{N}$ |
| entity index | $n$ | $1 .. N$ |
| set of entities | $E$ | $\{e_1, \ldots, e_n\}$ |
| spatial entity dimensionality | $D_x$ | $\mathbb{N}$ |
| entity feature dimensionality | $D_m$ | $\mathbb{N}$ |
| entity location (coordinate) | $\mathbf{x}$ | $\mathbb{R}^{D_x}$ |
| entity properties (entity features) | $\mathbf{m}$ | $\mathbb{R}^{D_m}$ |
| identifier representation dimensionality | $D_u$ | $\mathbb{N}$ |
| number of latent vectors | $L$ | $\mathbb{N}$ |
| latent vector dimensionality | $D_z$ | $\mathbb{N}$ |
| trajectory of a system (locations of entities over time) | $\mathbf{X}$ | $\mathbb{R}^{T_o \times N \times D_x}$ |
| entity locations at $t$ | $\mathbf{X}^t$ | $\mathbb{R}^{N \times D_x}$ |
| entity $i$ of trajectory at $t$ | $\mathbf{X}_i^t$ | $\mathbb{R}^{D_x}$ |
| trajectory in latent space | $\mathbf{Z}$ | $\mathbb{R}^{T_o \times L \times D_z}$ |
| latent system state at $t$ | $\mathbf{Z}^t$ | $\mathbb{R}^{L \times D_z}$ |
| time invariant features of entities | $\mathbf{M}$ | $\mathbb{R}^{N \times D_m}$ |
| matrix of identifier embeddings | $\mathbf{U}$ | $\mathbb{R}^{N \times D_u}$ |
| projection matrices | $\mathbf{Q}, \mathbf{K}, \mathbf{V}$ | not specified; depends on number of heads etc. |
| identifier assignment function | $\mathtt{ida}(\cdot)$ | $E \mapsto \mathcal{I}$ |
| encoder | $\mathcal{E}(.)$ | $\mathbb{R}^{N \times (D_u + D_x + D_m)} \mapsto \mathbb{R}^{L \times D_z}$ |
| decoder | $\mathcal{D}(.)$ | $\mathbb{R}^{L \times D_z} \times \mathbb{R}^{N \times D_u} \mapsto \mathbb{R}^{N \times (D_x + D_m)}$ |
| approximator (time dynamics model) | $\mathcal{A}(.)$ | $\mathbb{R}^{T \times L \times D_z} \mapsto \mathbb{R}^{T \times L \times D_z}$ |
| loss function | $\mathcal{L}(.,.)$ | var. |
| time parameter of the flow-based model | $\tau$ | $[0, 1]$ |
| noise distribution | $\mathbf{o}_0$ | $\mathbb{R}^{T \times L \times D_z}$ |
| de-noised de-masked trajectory | $\mathbf{o}_1 = \mathbf{Z}$ | $\mathbb{R}^{T \times L \times D_z}$ |
| flow-based model "velocity prediction" (neural net) | $\boldsymbol{v}_\theta(\mathbf{o}_\tau, \tau)$ | $\mathbb{R}^{T \times L \times D_z} \times \mathbb{R} \mapsto \mathbb{R}^{T \times L \times D_z}$ |
| flow-based model "data prediction" (neural net) | $\boldsymbol{o}_\theta(\mathbf{o}_\tau, \tau)$ | $\mathbb{R}^{T \times L \times D_z} \times \mathbb{R} \mapsto \mathbb{R}^{T \times L \times D_z}$ |
| neural network parameters | $\theta$ | undef. |

# B Architecture Details

## B.1 Architecture Overview in Detail

Figure 5 shows an expanded view of our architecture, and how the different components of LAM-SLIDE interact.

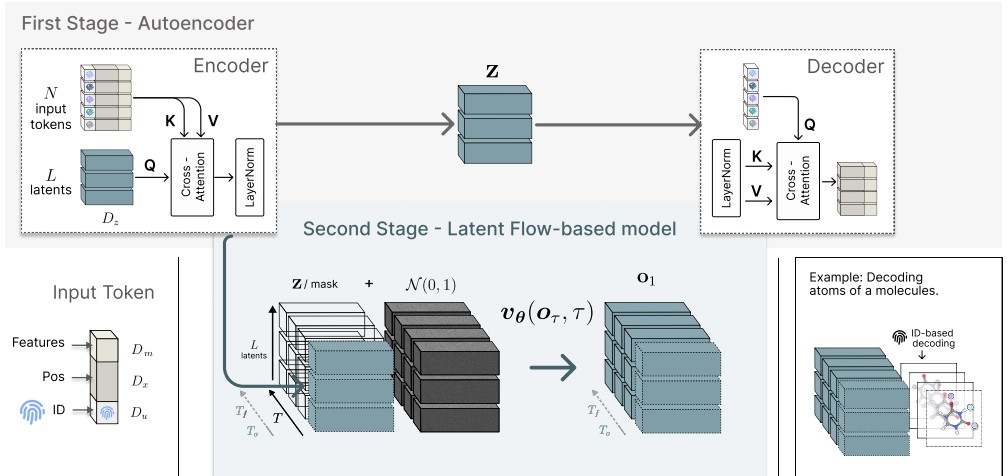

Figure 5: Expanded architectural overview. **First Stage**: The model is trained to reconstruct the encoded system by querying the latent system representation by IDs. **Second Stage**: Latent flow-based model is trained to predict multiple masked future timesteps. The predicted system states are decoded by the frozen decoder.

## B.2 Identifier Assignment

We provide pseudocode for the identifier creation in Algorithm 1. This algorithm prevents the reuse of already assigned IDs, maintaining unique IDs across all entities. From a practical perspective, we sample IDs randomly so that all entity embeddings receive gradient updates.

---

**Algorithm 1:** Identifier Construction

**Input** : number of entities $N$; identifier pool size $|\mathcal{I}|$ where $N \leqslant |\mathcal{I}|$; embedding dimension $D_u$
**Output** : $\mathbf{U} \in \mathbb{R}^{N \times D_u}$

1 $\mathbf{U} \leftarrow$ empty matrix of size $N \times D_u$
2 $S \leftarrow \{\}$                                                      // track assigned identifiers
3 **for** $i \leftarrow 1$ **to** $N$ **do**
4     $r \leftarrow \text{RandomSample}(\mathcal{I} \setminus S)$
5     $S \leftarrow S \cup \{r\}$
6     $\mathbf{e}_r \leftarrow \text{Embedding}(r)$                                   // learnable embeddings
7     $\mathbf{U}[i] \leftarrow \mathbf{e}_r$
8 **return** $\mathbf{U}$

---

## B.3 Encoder and Decoder

We provide pseudocode of the forward passes for encoding ($\mathcal{E}$) to and decoding ($\mathcal{D}$) from the latent system space of LAM-SLIDE in Algorithm 2 and Algorithm 3 respectively. In general, encoder and decoder blocks follow the standard Transformer architecture [91] with feedforward and normalization layers. To simplify the explanation, we omitted additional implementation details here and refer readers to our provided source code.

**Algorithm 2:** Encoder Function $\mathcal{E}$ (Cross-Attention)

**Input** : input data $\mathbf{XMU} = [\mathbf{X}, \mathbf{M}, \mathbf{U}] \in \mathbb{R}^{N \times (D_x + D_m + D_u)}$
**Output** : latent system state $\mathbf{Z} \in \mathbb{R}^{L \times D_z}$
**Internal parameters** : learned latent queries $\mathbf{Z}_{\text{init}} \in \mathbb{R}^{L \times D_z}$

1  $\mathbf{K} \leftarrow \text{Linear}(\mathbf{XMU})$
2  $\mathbf{V} \leftarrow \text{Linear}(\mathbf{XMU})$
3  $\mathbf{Q} \leftarrow \text{Linear}(\mathbf{Z}_{\text{init}})$
4  **return** $\text{LayerNorm}\big(\text{Attention}(\mathbf{Q}, \mathbf{K}, \mathbf{V})\big)$  `// without learnable affine parameters`

---

**Algorithm 3:** Decoder Function $\mathcal{D}$ (Cross-Attention)

**Input** : latent system representation $\mathbf{Z} \in \mathbb{R}^{L \times D_z}$; entity representation $\mathbf{u} \in \mathbb{R}^{D_u}$ drawn from $\mathbf{U} \in \mathbb{R}^{N \times D_u}$
**Output** : $[\mathbf{x}, \mathbf{m}] \in \mathbb{R}^{D_x + D_m}$

1  $\mathbf{Z} \leftarrow \text{LayerNorm}(\mathbf{Z})$                    `// without learnable affine parameters`
2  $\mathbf{K} \leftarrow \text{Linear}(\mathbf{Z})$
3  $\mathbf{V} \leftarrow \text{Linear}(\mathbf{Z})$
4  $\mathbf{q} \leftarrow \text{Linear}(\mathbf{u})$
5  **return** $\text{Attention}\big([\mathbf{q}], \mathbf{K}, \mathbf{V}\big)$

---

For the decoding functionality presented in Algorithm 3, we made use of multiple specific decoder blocks depending on the actual task (e.g., for the molecules dataset, we use one decoder block for atom positions and one decoder block for atom types).

## B.4  Latent Flow Model

We provide pseudocode of the data prediction network $\boldsymbol{o}_\theta$ forward pass in Algorithm 4. The latent layer functionality is given by Algorithm 5. The architecture of the latent layers (i.e., our flow model) is based on Dehghani et al. [23], with the additional usage of adaptive layer norm (adaLN) [69] as also used for Diffusion Transformers [67]. The implementation is based on ParallelMLP block code from Black Forest Labs [14], which was adapted to use it along the latent dimension as well as along the temporal dimension (see Figure 6). The velocity model is obtained via reparameterization as outlined in App. C.1.

---

**Algorithm 4:** Latent Flow Model $\boldsymbol{o}_\theta$ (data prediction network)

**Input** : noise-interpolated data $\mathbf{o}_{\text{inter}} \in \mathbb{R}^{T \times L \times D_z}$; diffusion time $\tau$ used for interpolation; conditioning $\mathbf{C} \in \mathbb{R}^{T \times L \times D_z}$; conditioning mask $\mathbf{B} \in \{0, 1\}^{T \times L \times D_z}$
**Output** : prediction of original data (not interpolated with noise) $\mathbf{o} \in \mathbb{R}^{T \times L \times D_z}$

1  $\boldsymbol{\tau} \leftarrow \text{Embed}(\tau)$
2  $\mathbf{o} \leftarrow \text{Linear}(\mathbf{o}_{\text{inter}}) + \text{Linear}(\mathbf{C}) + \text{Embed}(\mathbf{B})$
3  **for** $i \leftarrow 1$ **to** *num_layers* **do**
4  $\quad \mathbf{o} \leftarrow \text{LatentLayer}(\mathbf{o}, \boldsymbol{\tau})$
5  $\alpha, \beta, \gamma \leftarrow \text{Linear}(\text{SiLU}(\boldsymbol{\tau}))$
6  **return** $\mathbf{o} + \gamma \odot \text{MLP}\big(\alpha \odot \text{LayerNorm}(\mathbf{o}) + \beta\big)$

---

**Algorithm 5:** LatentLayer

**Input** : $\mathbf{o} \in \mathbb{R}^{T \times L \times C}$; diffusion time embedding $\boldsymbol{\tau}$
**Output** : updated $\mathbf{o} \in \mathbb{R}^{T \times L \times C}$

1  $\mathbf{o} += \text{ParallelMLPAttentionWithRoPE}(\mathbf{o}, \boldsymbol{\tau}, \dim = 0)$
2  $\mathbf{o} += \text{ParallelMLPAttentionWithRoPE}(\mathbf{o}, \boldsymbol{\tau}, \dim = 1)$
3  **return** $\mathbf{o}$

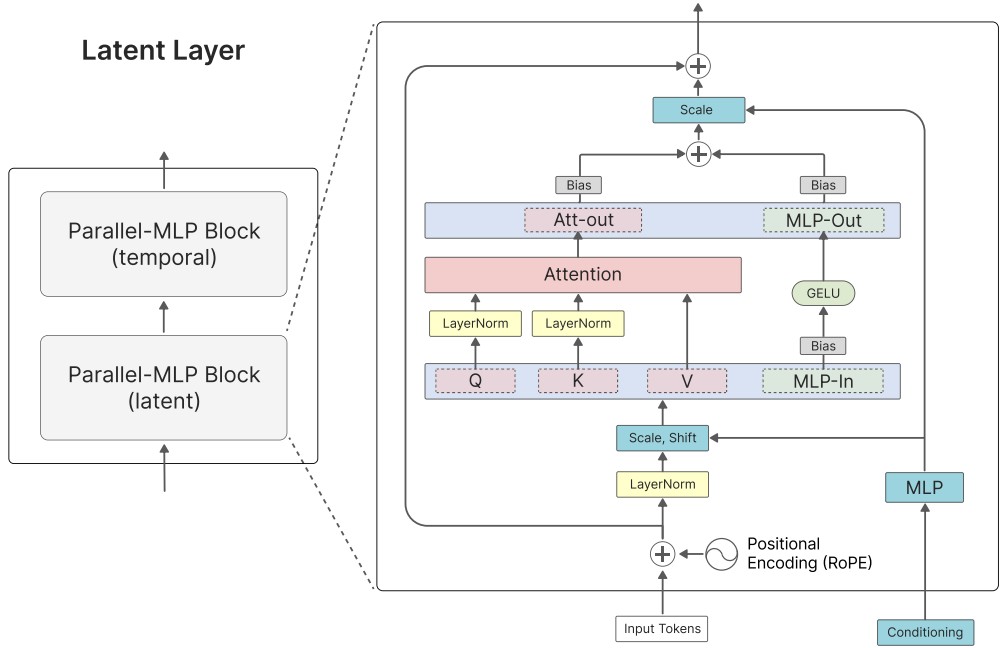

Figure 6: **Left**: LatentLayer of our method, consisting of a latent and a temporal ParallelMLP block. **Right**: Zoomed in view of the ParalellMLP block

Using `einops` [76] notation, the latent layer in Figure 6 can be expressed as:

$$\mathbf{o}' \leftarrow \texttt{rearrange}(\mathbf{o}, \ (B \ L) \ T \ D_z \rightarrow (B \ T) \ L \ D_z)$$
$$\mathbf{o}' \leftarrow l^i_\psi(\mathbf{o}', \tau)$$
$$\mathbf{o}' \leftarrow \texttt{rearrange}(\mathbf{o}', \ (B \ T) \ L \ D_z \rightarrow (B \ L) \ T \ D_z)$$
$$\mathbf{o}' \leftarrow l^i_\phi(\mathbf{o}', \tau)$$

with parameters sets $\psi$ and $\phi$, where for the latent block, the time dimension gets absorbed into the batch dimension, and for the temporal block, the latent dimension gets absorbed into the batch dimension.

## B.5 Python Pseudocode

This section shows Python pseudocode for training and inference for the latent approximator model.

```python
def latent_layer(z, t, z_cond):
    ## Latent layer of the approximator (high level)
    z = z + embed_cond(z_cond)
    t_embed = embed_time(t)

    # T x L x Dz, attention is calculated over L    | ref [4,5]
    z = attention_spatial(z, t_embed)

    z = rearrange(z, "T L Dz -> L T Dz")

    # L x T x Dz, attention is calculated over T    | ref [4,5]
    z = attention_temporal(z, t_embed)

    z = rearrange(z, "L T Dz -> T L Dz")

    return z
```

```python
#################
### Training ####
#################

# T x N x Dn (N=number of entities, D=[x,y,z,atom_type])
# e.g. Aspirin 21x4
x
ids
# T binary mask [1, 0, 0, ...], e.g., one conditioning frame
mask

# Encoding and conditioning setup

# T x L x Dz Encoder state into latent vector
z1 = encode(x, ids)

# T x L x Dz Contains only the conditioning frames,
# the other frames are masked out and set to 0.
z_cond = mask_frames(z1, mask)

# Sample z0 from a normal distribtion
z0 = torch.randn_like(z1)

# Sample t uniformly
t = torch.uniform(0, 1)

# Create interpolation
zt = t * z1 + (1 - t) * z0

# Loss
z1_hat = approximator(zt, t, z_cond)
loss = torch.mean((z1_hat - z1)**2)
loss.backward()

#################
### Inference ####
#################

# T x N x Dn, but non-conditioning frames are zero
x_cond

# Transform model trained on data prediction into velocity
# prediction model.
velocity_model = data_to_velocity_model(approximator)

# T x L x Dz, contains only the conditioning
# frames/ missing frames are set to 0
z_cond = encode(x_cond, ids)
z0 = torch.randn_like(z1)

# Solve ODE torchdiffeq (https://github.com/rtqichen/torchdiffeq)
z1_pred = solver(velocity_model, z0, z_cond)

x_hat = decoder(z1_pred)
```

# C Additional Information on Stochastic Interpolants

**General interpolants**. The stochastic interpolants framework [2, 3, 60] is defined without reference to a forward SDE, which allows a lot of flexibility; any choice of $\alpha_t$ and $\sigma_t$, satisfying the following conditions, is possible:

1. $\alpha_\tau^2 + \sigma_\tau^2 > 0$;
2. $\alpha_\tau$ and $\sigma_\tau$ are differentiable for all $\tau \in [0, 1]$
3. $\alpha_0 = \sigma_1 = 0$ and $\alpha_1 = \sigma_0 = 1$

**Interpolants**. Two common choices for $\alpha_t$ and $\sigma_t$ are the Linear and a Generalized Variance-Preserving (GVP) path:

$$\text{Linear:} \quad \alpha_\tau = \tau, \qquad\qquad\qquad \sigma_\tau = 1 - \tau \tag{7}$$

$$\text{GVP:} \quad \alpha_\tau = \sin\left(\frac{1}{2}\pi\tau\right), \qquad\qquad \sigma_\tau = \cos\left(\frac{1}{2}\pi\tau\right) \tag{8}$$

## C.1 Parametrization

Our latent flow-based model is implemented via data prediction objective [55, 48], with the aim to have small differences:

$$||\hat{\boldsymbol{o}}_\theta(\mathbf{o}; \tau) - \mathbf{o}_1||^2 . \tag{9}$$

The velocity model $\hat{\boldsymbol{v}}_\theta$ is obtained by reparameterization according to Lipman et al. [55]:

$$\hat{\boldsymbol{v}}_\theta(\mathbf{o}, \tau) = \hat{\boldsymbol{s}}_\theta(\mathbf{o}; \tau)\left(\frac{\dot{\alpha}_\tau \sigma_\tau^2}{\alpha_\tau} - \sigma_\tau \dot{\sigma}_\tau\right) + \frac{\dot{\alpha}_\tau}{\alpha_\tau}\mathbf{o} \tag{10}$$

where

$$\hat{\boldsymbol{s}}_\theta(\mathbf{o}; \tau) = -\sigma_\tau^{-2}(\mathbf{o} - \alpha_\tau \hat{\boldsymbol{o}}_\theta(\mathbf{o}; \tau)) . \tag{11}$$

For integration, we employed the `torchdiffeq` package [20], which provides solvers for differential equations.

# D  Additional related work

## D.1  Molecular Dynamics (MD)

The most fundamental concepts used to describe the dynamics of molecules nowadays are those given by the laws of quantum mechanics. The SchrÃűdinger equation is a partial differential equation that gives the evolution of the complex-valued wave function $\psi$ over time $t$: $i\hbar\frac{\partial\psi}{\partial t} = \hat{H}(t)\psi$. Here $i$ is the imaginary unit with $i^2 = -1$, $\hbar$ is the reduced Planck constant, and $\hat{H}(t)$ is the Hamiltonian operator at time $t$, which is applied to a function $\psi$ and maps to another function. It determines how a quantum system evolves with time, and its eigenvalues correspond to measurable energy values of the quantum system. The solution to SchrÃűdinger's equation in the many-body case (particles $1, \ldots, N$) is the wave function $\psi(\mathbf{x}_1, \ldots, \mathbf{x}_N, t) : \bigtimes_{i=1}^{N} \mathbb{R}^3 \times \mathbb{R} \to \mathbb{C}$ which we abbreviate as $\psi(\{\mathbf{x}\}, t)$. It's the square modulus $|\psi(\{\mathbf{x}\}, t)|^2 = \psi^*(\{\mathbf{x}\}, t)\psi(\{\mathbf{x}\}, t)$ is usually interpreted as a probability density to measure the positions $\mathbf{x}_1, \ldots, \mathbf{x}_N$ at time $t$, whereby the normalization condition $\int \ldots \int |\psi(\{\mathbf{x}\}, t)|^2 d\mathbf{x}_1 \ldots d\mathbf{x}_N = 1$ holds for the wave function $\psi$.

Analytic solutions of $\psi$ for specific operators $\hat{H}(t)$ are hardly known and are only available for simple systems like free particles or hydrogen atoms. In contrast to that are proteins, which consist of many thousands of atoms. However, already for much smaller quantum systems, approximations are needed. A famous example is the BornâĂŞOppenheimer approximation, where the wave function of the multi-body system is decomposed into parts for heavier atom nuclei and the light-weight electrons, which usually move much faster. In this case, one obtains a SchrÃűdinger equation for the electron's movement and another SchrÃűdinger equation for the nucleus's movement. A much faster option than solving a second SchrÃűdinger equation for the motion of the nuclei is to use the laws from classical Newtonian dynamics. The solution of the first SchrÃűdinger equation defines an energy potential, which can be utilized to obtain forces $\mathbf{F}_i$ on the nuclei and to update nuclei positions according to Newton's equation of motion: $\mathbf{F}_i = m_i \ddot{\mathbf{q}}_i(t)$ (with $m_i$ being the mass of particle $i$ and $\mathbf{q}_i(t)$ describing the motion trajectory of particle $i$ over time $t$).

Additional complexity in studying molecule dynamics is introduced by the environmental conditions surrounding molecules. Maybe the most important property is temperature. For biomolecules, it is often of interest to assume that they are dissolved in water. To model temperature, a usual strategy is to assume a system of coupled harmonic oscillators to model a heat bath, from which Langevin dynamics can be derived [29, 102]. The investigation of the relationship between quantum-mechanical modeling of heat baths and Langevin dynamics remains a current research topic, with various aspects, including the coupling of oscillators and the introduction of Markovian properties with stochastic forces. For instance, Hoel & Szepessy [39] studies how canonical quantum observables are approximated by molecular dynamics. This includes the definition of density operators, which behave according to the quantum Liouville-von Neumann equation.

The forces in molecules are usually given as the negative derivative of the (potential) energy: $\mathbf{F}_i = -\nabla E$. In the context of molecules, $E$ is usually assumed to be defined by a force field, which is a parameterized sum of intra- and intermolecular interaction terms. An example is the Amber force field [72, 18]:

$$E = \sum_{\text{bonds } r} k_b(r - r_0)^2 + \sum_{\text{angles } \theta} k_\theta(\theta - \theta_0)^2 + \tag{12}$$

$$\sum_{\text{dihedrals } \phi} V_n(1 + cos(n\phi - \gamma)) + \sum_{i=1}^{N-1} \sum_{j=i+1}^{N} \left( \frac{A_{ij}}{R_{ij}^{12}} - \frac{B_{ij}}{R_{ij}^{6}} + \frac{q_i q_j}{\epsilon R_{ij}} \right)$$

Here $k_b, r_0, k_\theta, \theta_0, V_n, \gamma, A_{ij}, B_{ij}, \epsilon, q_i, q_j$ serve as force field parameters, which are found either empirically or which might be inspired by theory.

Newton's equations of motion for all particles under consideration form a system of ordinary differential equations (ODEs), to which various numerical integration schemes, such as Euler, Leapfrog, or Verlet, can be applied to obtain particle position trajectories for given initial positions and velocities. If temperature is included, the resulting Langevin equations form a system of stochastic differential equations (SDEs), and Langevin integrators can be employed. It should be mentioned

that it is often necessary to use very small integration timesteps to avoid large approximation errors. This, however, increases the time needed to find new stable molecular configurations.

## D.2 Relationship to Graph Foundation Models

From our perspective, LAM-SLIDE bears a relationship to graph foundation models [GFMs; 56, 62]. Bommasani et al. [16] consider foundation models to be *trained on broad data at scale* and to be *adaptable to a wide range of downstream tasks*. Mao et al. [62] argue that graphs are more diverse than natural language or images, and therefore, there are unique challenges for GFMs. Especially, they mention that *none of the current GFM have the capability to transfer across all graph tasks and datasets from all domains*. It is for sure true that LAM-SLIDE is not a GFM in this sense. However, it might be debatable whether LAM-SLIDE might serve as a domain- or task-specific GFM. While we primarily focused on a trajectory prediction task and are, from that point of view, task-specific, we observed that our trained models can generalize across different molecules or scenes, which may seem quite remarkable given that it is common practice to train specific trajectory prediction models for individual molecules or scenes. Nevertheless, it was not our aim in this research to provide a GFM, as we believe that this would require further investigation into additional domains and could also necessitate, for instance, examining whether emergent abilities might arise with larger models and more training data [56].

## D.3 Relationship to Video and Language Diffusion Models

We want to elaborate our perspective on the relationship between LAM-SLIDE and recent advances in video [15] and language diffusion models [79, 59]. At their core, these approaches share a fundamental similarity: they can be conceptualized as a form of unmasking.

In video diffusion models, the model unmasks future frames; in language diffusion models, the model unmasks unknown tokens. Both paradigms learn to recover information that is initially obscured in the sequence, and importantly, both methods do so in parallel over the entire input sequence [9], compared to autoregressive models, which predict a single frame or token at a time.

Similarly, LAM-SLIDE represents each timestep as a set of latent tokens (or as a single token when concatenated). This perspective allows us to seamlessly incorporate recent advances from both video and language diffusion research into our modeling paradigm.

# E Experimental Details

## E.1 Datasets

**Pedestrian Movement**. The pedestrian movement dataset, along with its data processing, is available at `https://github.com/MediaBrain-SJTU/EqMotion`.

**Basketball Player Movement**. The dataset, along with its predefined splits, is available at `https://github.com/xupei0610/SocialVAE`. Data processing is provided in our source code.

**N-Body**. The dataset creation scripts, along with their predefined splits, are available at `https://github.com/hanjq17/GeoTDM`.

**Small Molecules (MD17)**. The MD17 dataset is available at `http://www.sgdml.org/#datasets`. Preprocessing and dataset splits follow Han et al. [35] and can be accessed through their GitHub repository at `https://github.com/hanjq17/GeoTDM`. The dataset comprises 5,000 training, 1000 validation, and 1000 test trajectories for each molecule.

**Tetrapeptides**. The dataset, including the full simulation parameters for ground truth simulations, is sourced from Jing et al. [44] and is publicly available in their GitHub repository at `https://github.com/bjing2016/mdgen`. The dataset comprises 3,109 training, 100 validation, and 100 test peptides.

## E.2 Condition and Prediction Horizon

Table 7 shows the conditioning and prediction horizon for the individual experiments. For the Tetrapeptides experiments, we predicted 1000 steps in parallel and reconditioned the model ten times on the last frame for each predicted block, this concept is similar to Arriola et al. [9].

Table 7: Number of conditioning and predicted frames for the different experiments.

| Experiment | Conditioning Frames | Predicted Frames | Total Frames |
|---|---|---|---|
| Pedestrian trajectory forecasting (ETH-UCY) | 8 | 12 | 20 |
| Basketball (NBA) | 8 | 12 | 20 |
| N-Body | 10 | 20 | 30 |
| Molecular Dynamics (MD17) | 10 | 20 | 30 |
| Molecular Dynamics - Tetrapeptides (4AA) | 1 | 9 999 | 10 000 |

## E.3 Implementation Details

**Training procedure**. (i) **First Stage**. In the first stage, we train the encoding and decoding functions $\mathcal{E}$ and $\mathcal{D}$ in an auto-encoding fashion, i.e., we optimize for a precise reconstruction of the original system state representation from its latent representation. For discrete features (e.g., atom type, residue type), we tend to use a cross-entropy loss, whereas for continuous features we use a regression loss (e.g., position, distance). The loss functions for each task are summarized in Appendix E.5. Notably, the entity identifier assignment is also random. (ii) **Second Stage**. In the second stage, we freeze the encoder and train the approximator to model the temporal dynamics via the encoded latent system representations. To learn a consistent behavior over time, we pass **U** from the encoder $\mathcal{E}$ to the decoder $\mathcal{D}$. To avoid high variance latent spaces, we used layer-normalization [10] (see Appendix E.3).

**Data Augmentation**. To compensate for the absence of built-in inductive biases such as equivariance/invariance with respect to spatial transformations, we apply random rotations and translations to the input coordinates.

**Identifiers**. For the embedding of the identifiers we use a `torch.nn.Embedding` [66] layer, where we assign a random subset of the possible embeddings to the entities in each training step. See also Algorithm 1.

**Latent space regularization**. To avoid high variance latent spaces, Rombach et al. [77] relies on KL-reg., imposing a small KL-penalty towards a standard normal on the latent space, as used in VAE [49]. Recent work [100] has shown that layer normalization [10] can achieve similar regulatory effects without requiring an additional loss term and simplifying the training procedure. We adapt this approach in our method (see the left part of Figure 2).

**Latent Model**. For the latent Flow Model we additionally apply auxiliary losses for the individual tasks, as shown in App. E.5. Where we decode the predicted latent system representations and back-propagate through the frozen decoder to the latent model.

**MD17**. We train a single model on all molecules – a feat that is structurally encouraged by the design of LaM-Slide . For ablation, we also train GeoTDM [35] on all molecules and evaluate the performance on each one of them ("all→each" in the Table 13). Interestingly, we also observe consistent improvements in the GeoTDM performance. However, GeoTDM's performance does not reach that of LaM-Slide .

**Tetrapeptides**. For the experiments on tetrapeptides in Section 4.5, we employ the Atom14 representation as used in AlphaFold [1]. In this representation, each entity corresponds to one amino acid of the tetrapeptide, where multiple atomic positions are encoded into a single vector of dimension $D_x = 3 \times 14$. Masked atomic positions are excluded from gradient computation during model updates. This representation is computationally more efficient.

## E.4 Loss Functions

This section defines the losses, which we use throughout training:

**Position Loss**.

$$\mathcal{L}_{\text{pos}}(\mathbf{X}^t, \hat{\mathbf{X}}^t) = \frac{1}{N} \sum_{i=1}^{N} ||\mathbf{X}_i^t - \hat{\mathbf{X}}_i^t||_2^2 \tag{13}$$

**Inter-distance Loss**.

$$\mathcal{L}_{\text{int}}(\mathbf{X}^t, \hat{\mathbf{X}}^t) = \frac{1}{N^2} \sum_{i=1}^{N} \sum_{j=1}^{N} (D_{ij}(\mathbf{X}^t) - D_{ij}(\hat{\mathbf{X}}^t))^2 \tag{14}$$

with

$$D_{ij}(\mathbf{X}^t) = ||\mathbf{X}_i^t - \mathbf{X}_j^t||_2 \tag{15}$$

**Cross-Entropy Loss**. Depending on the experiment, we have different CE losses depending on the problem, see App. E.5.

$$\mathcal{L}_{CE} = \frac{1}{N} \left( - \sum_{k=1}^{K} y_k \log(p_k) \right) \tag{16}$$

**Frame and Torsion Loss**. For the Tetrapeptide experiments, we employ two additional auxiliary loss functions tailored to better capture unique geometric constraints of proteins, complementing our primary optimization objectives: a frame loss $\mathcal{L}_{frame}$, which is based on representing all atoms withing a local reference frame [1, Algorithm 29], and a torsion loss torsion loss $\mathcal{L}_{\text{tors}}$ inspired by Jumper et al. [46].

## E.5 Hyperparameters

Tabs. 8 to 12 show the hyperparameters for the individual tasks, loss functions are as defined in App. E.4. For all trained models, we use the AdamW [50, 58] optimizer and use EMA [31] in each update step with a decay parameter of $\beta = 0.999$.

### E.6 Evaluation Details

**Tetrapeptides**. Our analysis of the Tetrapeptide trajectories utilized PyEMMA [84] and followed the procedure as Jing et al. [44], incorporating both Time-lagged Independent Component Analysis (TICA) [70] and Markov State Models (MSM) [42]. For the evaluation of these metrics, we relied on the implementations provided by [44].

### E.7 Computational Resources

Our experiments were conducted using a system with 128 CPU cores and 2048GB of system memory. Model training was performed on 4 NVIDIA H200 GPUs, each equipped with 140GB of VRAM. In total, roughly 5000 GPU hours were used in this work.

### E.8 Software

We used `PyTorch 2` [7] for the implementation of our models. Our training pipeline was structured with `PyTorch Lightning` [28]. We used `Hydra` [96] to run our experiments with different hyperparameter settings. Our experiments were tracked with `Weights & Biases` [12].

Table 8: Hyperparameter configuration for the pedestrian movement experiments (Section 4.1).

| First Stage | |
|---|---|
| **Network** | |
| **Encoder** | |
| Number of latents $L$ | 2 |
| Number of entity embeddings | 8 |
| Number of attention heads | 2 |
| Number of cross attention layers | 1 |
| Dimension latents $D_z$ | 32 |
| Dimension entity embedding | 128 |
| Dimension attention head | 16 |
| **Decoder** | |
| Number of attention heads | 2 |
| Number of cross attention layers | 1 |
| Dimension attention head | 16 |
| **Loss** | **Weight** |
| $\mathcal{L}_{pos}(\mathbf{X}, \hat{\mathbf{X}})$ | 1 |
| $\mathcal{L}_{int}(\mathbf{X}, \hat{\mathbf{X}})$ | 1 |
| **Training** | |
| Learning rate | 1e-4 |
| Learning rate scheduler | CosineAnnealing(min_lr=1e-7) |
| Batch size | 256 |
| Epochs | 3K |
| Precision | 32-Full |
| Batch size | 1024 |
| **Second Stage** | |
| **Setup** | |
| Condition | 8 Frames |
| Prediction | 12 Frames |
| **Network** | |
| Hidden dimension | 128 |
| Number of Layers | 6 |
| **Auxiliary - Loss** | **Weight** |
| $\mathcal{L}_{pos}(\mathbf{X}, \hat{\mathbf{X}})$ | 0.25 |
| $\mathcal{L}_{int}(\mathbf{X}, \hat{\mathbf{X}})$ | 0.25 |
| **Training** | |
| Learning rate | 1e-3 |
| Batch size | 64 |
| Epochs | 1K |
| Precision | BF16-Mixed |
| **Inference** | |
| Integrator | Euler |
| ODE steps | 10 |

Table 9: Hyperparameter configuration for the basketball player movement experiments (Section 4.2).

| First Stage | |
|---|---|
| **Network** | |
| **Encoder** | |
| Number of latents $L$ | 32 |
| Number of entity embeddings | 11 |
| Number of attention heads | 2 |
| Number of cross attention layers | 1 |
| Dimension latents $D_z$ | 32 |
| Dimension entity embedding | 128 |
| Dimension attention head | 16 |
| **Decoder** | |
| Latent dimension | 32 |
| Number of attention heads | 8 |
| Number of cross attention layers | 1 |
| **Loss** | **Weight** |
| $\mathcal{L}_{pos}(\mathbf{X}, \hat{\mathbf{X}})$ | 1 |
| $\mathcal{L}_{int}(\mathbf{X}, \hat{\mathbf{X}})$ | 1 |
| $\mathcal{L}_{CE}(\cdot, \cdot) - \text{Group}$ | 0.01 |
| $\mathcal{L}_{CE}(\cdot, \cdot) - \text{Team}$ | 0.01 |
| **Training** | |
| Learning rate | 1e-4 |
| Learning rate scheduler | CosineAnnealing(min_lr=1e-7) |
| Optimizer | AdamW |
| Batch size | 16 |
| **Second Stage** | |
| **Setup** | |
| Condition | 8 Frames |
| Prediction | 12 Frames |
| **Network** | |
| Hidden dimension $H$ | 128 |
| Number of Layers | 6 |
| **Auxiliary - Loss** | **Weight** |
| $\mathcal{L}_{pos}(\mathbf{X}, \hat{\mathbf{X}})$ | 0.25 |
| $\mathcal{L}_{int}(\mathbf{X}, \hat{\mathbf{X}})$ | 0.25 |
| **Training** | |
| Learning rate | 1e-3 |
| Learning rate scheduler | CosineAnnealing(min_lr=1e-7) |
| Batch size | 64 |
| Epochs | 500 |
| Precision | BF16-Mixed |
| **Inference** | |
| Integrator | Euler |
| ODE steps | 10 |

Table 10: Hyperparameter configuration for the N-Body experiments (Section 4.3).

| First Stage | |
| --- | --- |
| **Network** | |
| **Encoder** | |
| Number of latents $L$ | 16 |
| Number of entity embeddings | 10 |
| Number of attention heads | 2 |
| Number of cross attention layers | 1 |
| Dimension latents $D_z$ | 32 |
| Dimension entity embedding | 128 |
| Dimension attention head | 16 |
| **Decoder** | |
| Number of cross attention layers | 1 |
| Number of attention heads | 2 |
| Number of cross attention layers | 16 |
| **Loss** | **Weight** |
| $\mathcal{L}_{pos}(\mathbf{X}, \hat{\mathbf{X}})$ | 1 |
| $\mathcal{L}_{int}(\mathbf{X}, \hat{\mathbf{X}})$ | 1 |
| **Training** | |
| Learning rate | 1e-3 |
| Batch size | 128 |
| Epochs | 2K |
| Precision | 32-Full |
| **Second Stage** | |
| **Setup** | |
| Condition | 10 Frames |
| Prediction | 20 Frames |
| **Network** | |
| Hidden dimension | 256 |
| Number of Layers | 6 |
| **Auxiliary - Loss** | **Weight** |
| $\mathcal{L}_{pos}(\mathbf{X}, \hat{\mathbf{X}})$ | 0.0 |
| $\mathcal{L}_{int}(\mathbf{X}, \hat{\mathbf{X}})$ | 0.0 |
| **Training** | |
| Learning rate | 1e-3 |
| Learning rate scheduler | CosineAnnealing(min_lr=1e-7) |
| Batch size | 64 |
| Epochs | 1K |
| Precision | BF16-Mixed |
| **Inference** | |
| Integrator | Euler |
| ODE steps | 10 |

Table 11: Hyperparameter configuration for the small molecule (MD17) experiments (Section 4.4).

| First Stage | |
| --- | --- |
| **Network** | |
| **Encoder** | |
| Number of latents $L$ | 32 |
| Number of entity embeddings | 8 |
| Number of attention heads | 2 |
| Number of cross attention layers | 1 |
| Dimension latents $D_z$ | 32 |
| Dimension entity embedding | 128 |
| Dimension attention head | 16 |
| **Decoder** | |
| Number of cross attention layers | 1 |
| Number of attention heads | 2 |
| Number of cross attention layers | 16 |
| **Loss** | **Weight** |
| $\mathcal{L}_{pos}(\mathbf{X}, \hat{\mathbf{X}})$ | 1 |
| $\mathcal{L}_{int}(\mathbf{X}, \hat{\mathbf{X}})$ | 1 |
| $\mathcal{L}_{CE}(\cdot, \cdot) -$ Atom type | 1 |
| **Training** | |
| Learning rate | 1e-4 |
| Batch size | 256 |
| Epochs | 3K |
| Precision | 32-Full |
| **Second Stage** | |
| **Setup** | |
| Condition | 10 Frames |
| Prediction | 20 Frames |
| **Network** | |
| Hidden dimension | 128 |
| Number of Layers | 6 |
| **Auxiliary - Loss** | **Weight** |
| $\mathcal{L}_{pos}(\mathbf{X}, \hat{\mathbf{X}})$ | 0.25 |
| $\mathcal{L}_{int}(\mathbf{X}, \hat{\mathbf{X}})$ | 0.25 |
| **Training** | |
| Learning rate | 1e-3 |
| Learning rate scheduler | CosineAnnealing(min_lr=1e-7) |
| Batch size | 64 |
| Epochs | 2K |
| Precision | BF16-Mixed |
| **Inference** | |
| Integrator | Euler |
| ODE steps | 10 |

Table 12: Hyperparameter configuration for the Tetrapeptides experiments (Section 4.5).

| First Stage | |
| --- | --- |
| **Network** | |
| **Encoder** | |
| Number of latents $L$ | 5 |
| Number of entity embeddings | 8 |
| Number of attention heads | 2 |
| Number of cross attention layers | 1 |
| Dimension latents $D_z$ | 96 |
| Dimension entity embedding | 128 |
| Dimension attention head | 16 |
| **Decoder** | |
| Number of attention heads | 2 |
| Number of cross attention layers | 1 |
| Dimension attention head | 16 |
| **Loss** | **Weight** |
| $\mathcal{L}_{pos}(\mathbf{X}, \hat{\mathbf{X}})$ | 1 |
| $\mathcal{L}_{int}(\mathbf{X}, \hat{\mathbf{X}})$ | 1 |
| $\mathcal{L}_{frame}(\mathbf{X}, \hat{\mathbf{X}})$ | 1 |
| $\mathcal{L}_{tors}(\mathbf{X}, \hat{\mathbf{X}})$ | 0.1 |
| $\mathcal{L}_{CE}(\cdot, \cdot)$ − Residue type | 0.001 |
| **Training** | |
| Learning rate | 1e-4 |
| Batch size | 16 |
| Epochs | 200K |
| Precision | 32-Full |
| **Second Stage** | |
| **Setup** | |
| Condition | 1 Frame |
| Prediction | 10,000 Frames (10x rollouts) |
| **Network** | |
| Hidden dimension | 384 |
| Number of Layers | 6 |
| **Auxiliary - Loss** | **Weight** |
| $\mathcal{L}_{pos}(\mathbf{X}, \hat{\mathbf{X}})$ | 0.25 |
| $\mathcal{L}_{int}(\mathbf{X}, \hat{\mathbf{X}})$ | 0.25 |
| $\mathcal{L}_{frame}(\mathbf{X}, \hat{\mathbf{X}})$ | 0.25 |
| **Training** | |
| Learning rate | 1e-3 |
| Optimizer | AdamW |
| Batch size | 64 |
| Epochs | 1.5K |
| Precision | BF16-Mixed |
| **Inference** | |
| Integrator | Dopri5 [20] |
| ODE steps | adaptive |

# F    Proofs

## F.1    Proof of Proposition 3.3

The proof is rather simple, but we include it for completeness.

**Proposition 3.3.** *Given an identifier pool $\mathcal{I}$ and a finite set of entities $E$, an identifier assignment pool I as defined by Definition 3.2 is non-empty if and only if $|E| \leqslant |\mathcal{I}|$.*

*Proof.* Assume $|E| > |\mathcal{I}|$. By pigeonhole principle, any function $f : E \mapsto \mathcal{I}$ must map at least two distinct elements of $E$ to the same element in $I$. Therefore, $f$ cannot be injective.

Conversely, if $|E| \leqslant |\mathcal{I}|$, we can construct an injective function from $E$ to $\mathcal{I}$ by assigning each element in $E$ a unique element in $\mathcal{I}$, which is possible because $\mathcal{I}$ has at least as many elements as $E$.

Therefore, an injective identifier assignment function $\mathtt{ida}(\cdot) \in I$ only exists if $|E| \leqslant |\mathcal{I}|$. Hence, the set I is non-empty in this case and empty otherwise.

## F.2    Proof of Proposition 3.4

**Proposition 3.4.** *Given an identifier pool $\mathcal{I}$ and a finite set of entities $E$ such that $|E| \leqslant |\mathcal{I}|$, the identifier assignment pool I as defined by Definition 3.2 contains finitely many injective functions.*

Let $n = |E|$ and $m = |\mathcal{I}|$, then the set of infective functions $I$ is bounded and finite:

$$|I| = (m-1)\dots(m-n+1) = \frac{m!}{(m-n)!} = (m)_n \leqslant \inf \tag{17}$$

Where $(m)_n$ is commonly referred to as *falling factorials*, the number of injective functions from a set of size $n$ to a set of size $m$.

# G Additional Experiments

To investigate the sensitivity of our model with respect to different hyperparameter settings, we conducted additional experiments.

## G.1 Number of Parameter

We conducted scaling experiments on both the MD17 and the Tetrapeptides (4AA) datasets to evaluate how LAM-SLIDE 's performance scales with model size. On MD17, we evaluate LAM-SLIDE using model variants with 1.7M, 2.1M, and 2.5M parameters. Our results show that, for nearly all molecules, performance improves with parameter count in terms of ADE/FDE, see Table 13. Similarly, on the Tetrapeptides dataset, we evaluate using model variants with 4M, 7M, 11M, and 28M parameters. All performance metrics show consistent improvement with increased model capacity, see Table 14. These findings indicate the favorable scaling behavior of our method.

Table 13: **Method comparison for forecasting MD trajectories of small molecules**. Compared methods predict atom positions for 20 frames, conditioned on 10 input frames. Results are reported in terms of ADE/FDE, averaged over 5 sampled trajectories.

| | Aspirin | | Benzene | | Ethanol | | Malonaldehyde | | Naphthalene | | Salicylic | | Toluene | | Uracil | |
|---|---|---|---|---|---|---|---|---|---|---|---|---|---|---|---|---|
| | ADE | FDE | ADE | FDE | ADE | FDE | ADE | FDE | ADE | FDE | ADE | FDE | ADE | FDE | ADE | FDE |
| RF [52][a] | 0.303 | 0.442 | 0.120 | 0.194 | 0.374 | 0.515 | 0.297 | 0.454 | 0.168 | 0.185 | 0.261 | 0.343 | 0.199 | 0.249 | 0.239 | 0.272 |
| TFN [89][a] | 0.133 | 0.268 | 0.024 | 0.049 | 0.201 | 0.414 | 0.184 | 0.386 | 0.072 | 0.098 | 0.115 | 0.223 | 0.090 | 0.150 | 0.090 | 0.159 |
| SE(3)-Tr. [30][a] | 0.294 | 0.556 | 0.027 | 0.056 | 0.188 | 0.359 | 0.214 | 0.456 | 0.069 | 0.103 | 0.189 | 0.312 | 0.108 | 0.184 | 0.107 | 0.196 |
| EGNN [82][a] | 0.267 | 0.564 | 0.024 | 0.042 | 0.268 | 0.401 | 0.393 | 0.958 | 0.095 | 0.133 | 0.159 | 0.348 | 0.207 | 0.294 | 0.154 | 0.282 |
| EqMotion [94][a] | 0.185 | 0.246 | 0.029 | 0.043 | 0.152 | 0.247 | 0.155 | 0.249 | 0.073 | _0.092_ | 0.110 | 0.151 | 0.097 | 0.129 | 0.088 | 0.116 |
| SVAE [95][a] | 0.301 | 0.428 | 0.114 | 0.133 | 0.387 | 0.505 | 0.287 | 0.430 | 0.124 | 0.135 | 0.122 | 0.142 | 0.145 | 0.171 | 0.145 | 0.156 |
| GeoTDM 1.9M[a] | 0.107 | 0.193 | _0.023_ | _0.039_ | 0.115 | 0.209 | 0.107 | 0.176 | 0.064 | 0.087 | 0.083 | 0.120 | 0.083 | 0.121 | 0.074 | _0.099_ |
| GeoTDM 1.9M (all→each) | _0.091_ | _0.164_ | 0.024 | 0.040 | _0.104_ | _0.191_ | _0.097_ | _0.164_ | _0.061_ | _0.092_ | _0.074_ | _0.114_ | _0.073_ | _0.112_ | _0.070_ | 0.102 |
| LAM-SLIDE 2.5M | **0.059** | **0.098** | **0.021** | **0.032** | **0.087** | **0.167** | **0.073** | **0.124** | **0.037** | **0.058** | **0.047** | **0.074** | **0.045** | **0.075** | **0.050** | **0.074** |
| LAM-SLIDE 2.1M | 0.064 | 0.104 | 0.023 | 0.033 | 0.097 | 0.182 | 0.084 | 0.141 | 0.044 | 0.067 | 0.053 | 0.081 | 0.054 | 0.086 | 0.054 | 0.079 |
| LAM-SLIDE 1.7M | 0.074 | 0.117 | 0.025 | 0.037 | 0.110 | 0.195 | 0.097 | 0.159 | 0.053 | 0.074 | 0.063 | 0.091 | 0.064 | 0.094 | 0.064 | 0.089 |

[a] Results from Han et al. [35].

Table 14: **Method comparison for predicting MD trajectories of tetrapeptides**. The columns denote the JSD between distributions of *torsion angles* (backbone (BB), side-chain (SC), and all angles), the TICA, the MSM metric, and the number of parameters.

| | Torsions | | | TICA | | MSM | Params | Time |
|---|---|---|---|---|---|---|---|---|
| | BB | SC | All | 0 | 0,1 joint | | (M) | |
| 100 ns[a] | .103 | .055 | .076 | .201 | .268 | .208 | | $\sim$ 3h |
| MDGen[a] | .130 | **.093** | **.109** | .230 | .316 | .235 | 34 | $\sim$ 60s |
| LAM-SLIDE | **.128** | 0.122 | 0.125 | **.227** | **.315** | **.224** | 28 | $\sim$ 53s |
| LAM-SLIDE | .152 | .151 | .152 | .239 | .331 | .226 | 11 | |
| LAM-SLIDE | .183 | .191 | .187 | .26 | .356 | .235 | 7 | |
| LAM-SLIDE | .284 | .331 | .311 | .339 | .461 | .237 | **4** | |

[a] Results from Jing et al. [44].

## G.2 Number of Function Evaluations (NFEs)

We conduct a comparative analysis on computational efficiency by measuring the number of function evaluations (NFEs) required to achieve the performance results reported in the main section of our publication. As shown in Table 15, our approach demonstrates remarkable efficiency compared to the previous state-of-the-art method, GeoTDM [35], across N-Body simulations, MD17 molecular dynamics, and ETH pedestrian motion forecasting experiments. Our model consistently requires significantly fewer NFEs than GeoTDM to reach comparable or superior performance levels.

It is worth noting that flow-based models generally require fewer NFEs compared to diffusion-based approaches, such as GeoTDM. However, this efficiency advantage does not come at the expense of performance quality [27]. Indeed, the relationship between NFEs and performance is not strictly monotonic, as demonstrated in other domains. For instance, Esser et al. [27] achieved optimal image generation results in terms of FID [36] with 25 NFEs, demonstrating that computational efficiency and high performance can be achieved simultaneously with properly designed architectures.

Furthermore, in the case of the Tetrapeptides (4AA) experiments shown in Section 4.5, we employ an adaptive step size solver to achieve the reported performance, which yields better results than an Euler solver. We use Dopri5 as implemented in the `torchdiffeq` package [20].

Table 15: Comparison of the number of function evaluations (NFEs) for LAM-SLIDE and GeoTDM.

| | N-Body | MD17 | ETH |
|---|---|---|---|
| GeoTDM [35][a] | 1000 | 1000 | 100 |
| LAM-SLIDE | **10** | **10** | **10** |

[a] Results from Han et al. [35].

### G.3 Number of Learned Latent Vectors

We conducted experiments to quantify the relationship between model performance and the number of latent vectors $L$ using the MD17 dataset. As shown in Table 16, performance increases with the number of latent vectors $L$. Of particular significance is the performance at $L = 21$, which corresponds to the maximum number of entities, allowing us to investigate whether this constitutes an upper bound on model capacity. Notably, performance continues to improve at $L = 32$, indicating that model capacity scales favorably even beyond the number of entities. At $L = 16$, representing a compressed latent representation, our model remains competitive with the second-best method, GeoTDM [35].

We further analyze whether the improvement resulting from increasing $L$ is due solely to the encoder-decoder's reconstruction performance. Figure 8 shows the reconstruction error for varying number of latent vectors $L$. Even with substantially fewer latent vectors than entities, the model achieves good reconstruction performance. This gap suggests that the performance gains from increasing $L$ are not due to improved reconstruction, but rather to the model's ability to leverage the enlarged latent space representation more effectively.

### G.4 Identifier Pool Size

We evaluate the impact of identifier pool size using the MD17 dataset. This dataset contains at most 21 atoms, so, in general, an identifier pool of $|\mathcal{I}| = 21$ would be sufficient. However, for the results shown in the main paper, we used $|\mathcal{I}| = 32$. To investigate the impact of a larger identifier pool $\mathcal{I}$, we conduct additional experiments by training multiple first-stage models with varying identifier pool sizes and report the reconstruction error measured by Euclidean distance. Figure 7 shows that the reconstruction error increases with the size of the identifier pool, since a larger pool results in fewer updates to each entity embedding during training.

### G.5 Identifier Assignment

To assess the impact of different ID assignments on reconstruction performance, we run our model five times with different random ID assignments and measure the standard deviation across these assignments in terms of reconstruction error in Å. Results are shown in Figures 7 and 8. The low standard deviation across different ID assignments demonstrates the robustness of our model with respect to random ID assignment.

Table 16: Model performance in terms of ADF/FDE with respect to **different number of latent vectors** $L$.

| | Aspirin | | Benzene | | Ethanol | | Malonaldehyde | | Naphthalene | | Salicylic | | Toluene | | Uracil | |
|---|---|---|---|---|---|---|---|---|---|---|---|---|---|---|---|---|
| | ADE | FDE | ADE | FDE | ADE | FDE | ADE | FDE | ADE | FDE | ADE | FDE | ADE | FDE | ADE | FDE |
| LaM-Slide ($L = 4$) | 0.354 | 0.483 | 0.213 | 0.264 | 0.420 | 0.541 | 0.366 | 0.537 | 0.210 | 0.223 | 0.253 | 0.286 | 0.316 | 0.418 | 0.243 | 0.275 |
| LaM-Slide ($L = 8$) | 0.234 | 0.315 | 0.114 | 0.135 | 0.242 | 0.361 | 0.201 | 0.300 | 0.157 | 0.163 | 0.179 | 0.201 | 0.206 | 0.250 | 0.158 | 0.180 |
| LaM-Slide ($L = 16$) | 0.099 | 0.146 | 0.039 | 0.049 | 0.108 | 0.187 | 0.095 | 0.149 | 0.070 | 0.089 | 0.075 | 0.101 | 0.073 | 0.103 | 0.071 | 0.093 |
| LaM-Slide ($L = 21$) | 0.078 | 0.118 | 0.031 | 0.041 | 0.097 | 0.175 | 0.082 | 0.135 | 0.054 | 0.074 | 0.059 | 0.085 | 0.057 | 0.085 | 0.059 | 0.083 |
| LaM-Slide ($L = 32$) | **0.059** | **0.098** | **0.021** | **0.032** | **0.087** | **0.167** | **0.073** | **0.124** | **0.037** | **0.058** | **0.047** | **0.074** | **0.045** | **0.075** | **0.050** | **0.074** |

## G.6 Identifier Latent Utilization

We investigate how identifiers are utilized by our model, addressing three key questions: Does each identifier exhibit a unique query pattern? How do these patterns vary with latent space dimensionality? Do different attention heads learn distinct patterns?

We examined decoder attention weights in overparameterized ($N < L$) and compressed ($N \geq L$) settings using models with $L = 16$ and $L = 32$ latent vectors trained on MD17. Our analysis focuses on Aspirin molecules ($N = 21$ atoms) across different conformations.

The results in Figs. 9 to 12 reveal two main findings. First, identical atom identifiers produce consistent attention patterns across different conformations, with each identifier learning a unique addressing scheme distributed over latent vectors. This suggests that the identifiers function as "retrieval mechanisms" independent of stored content.

Second, attention head utilization adapts to the compression ratio. The overparameterized model ($L = 32$) primarily uses a single head, acting like a hash table-like retrieval. The compressed model ($L = 16$) utilizes both heads with distinct patterns, likely to mitigate identifier collisions in the limited latent space.

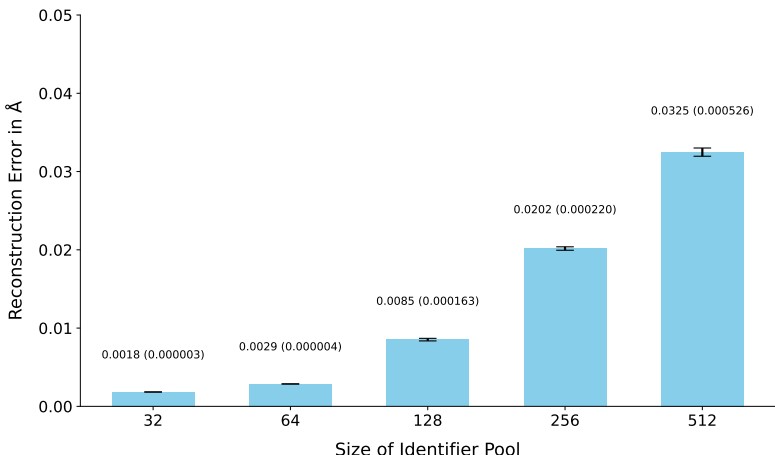

Figure 7: Reconstruction error in Å for the MD17 dataset. We report the reconstruction error for the encoder-decoder model for **different identifier pool sizes**. The error bars show the standard deviation across five runs with different random ID assignments.

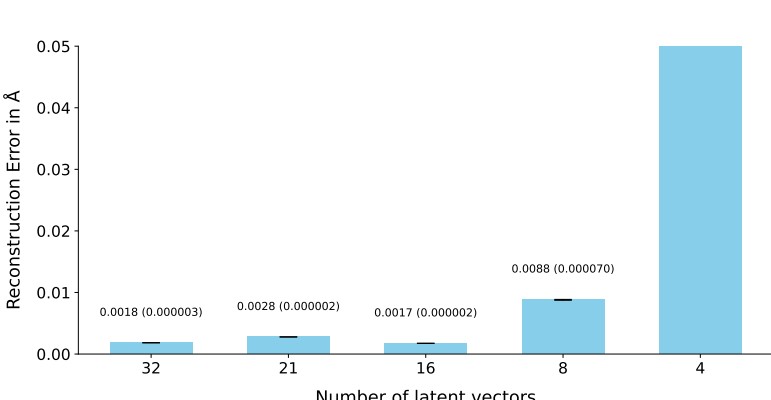

Figure 8: Reconstruction error in Å for the MD17 dataset. We report the reconstruction error for the encoder-decoder model for **different number of latent vectors** $L$. The error bars show the standard deviation across five runs with different random ID assignments.

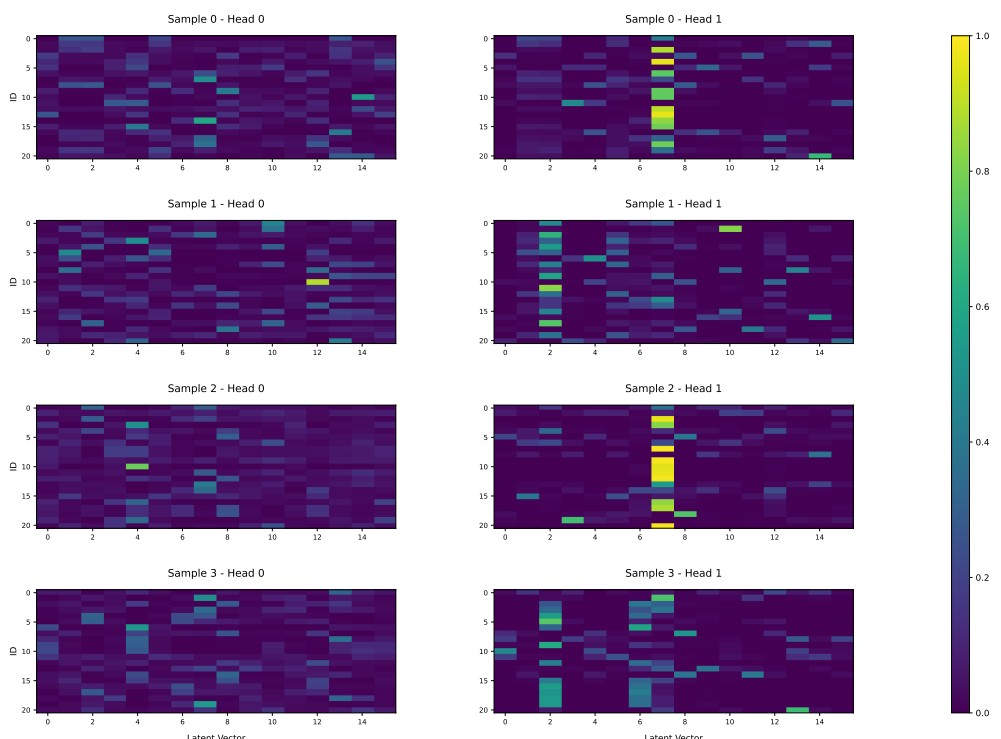

Figure 9: Attention patterns on the MD17 dataset for four random samples of the validation dataset, using a model trained with **16 latent vectors**. Each row corresponds to a single sample and displays the attention patterns across individual attention heads. Each sample employs a **distinct identifier assignment**.

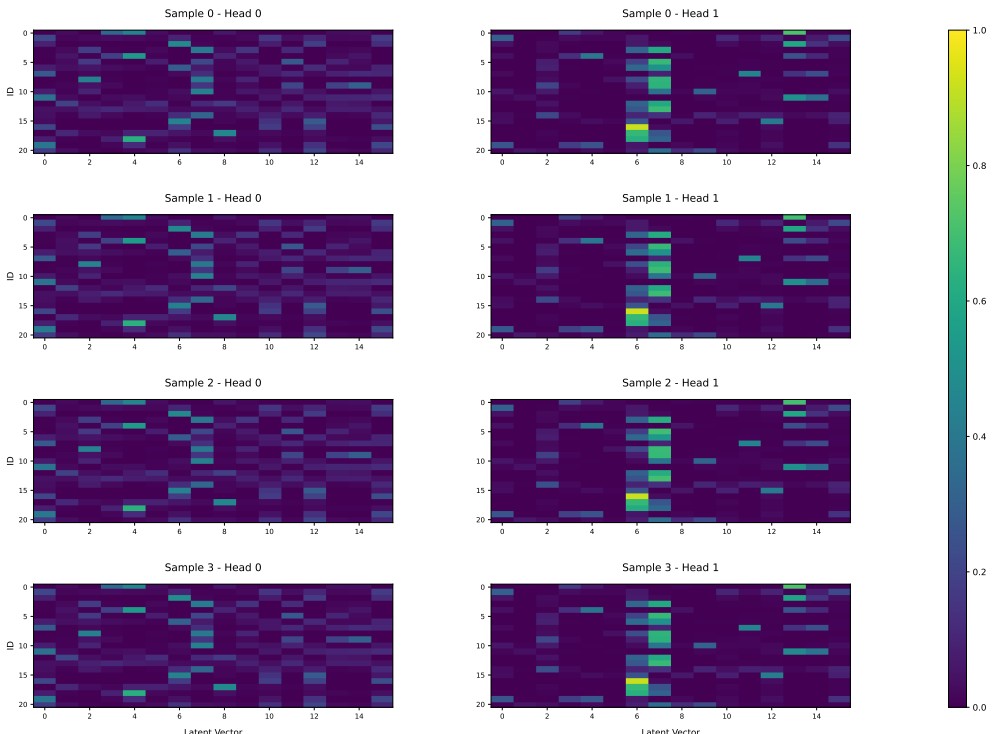

Figure 10: Attention patterns on the MD17 dataset for four random samples of the validation dataset, using a model trained with **16 latent vectors**. Each row corresponds to a single sample and displays the attention patterns across individual attention heads. Each sample employs the **same identifier assignment**.

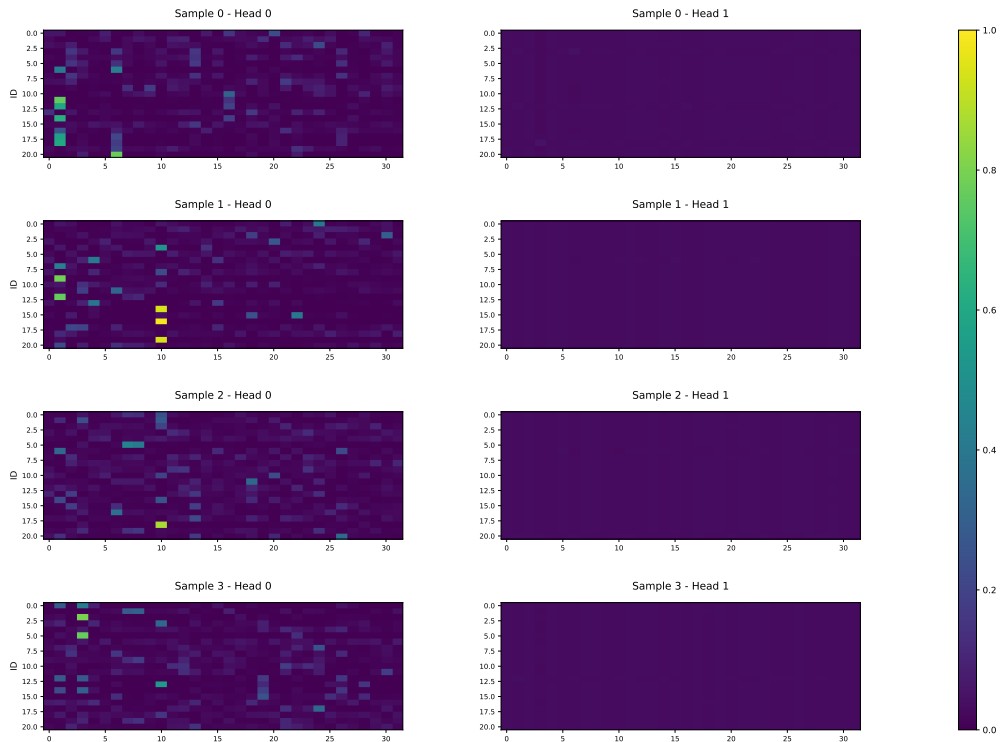

Figure 11: Attention patterns on the MD17 dataset for four random samples of the validation dataset, using a model trained with **32 latent vectors**. Each row corresponds to a single sample and displays the attention patterns across individual attention heads. Each sample employs a **distinct identifier assignment**.

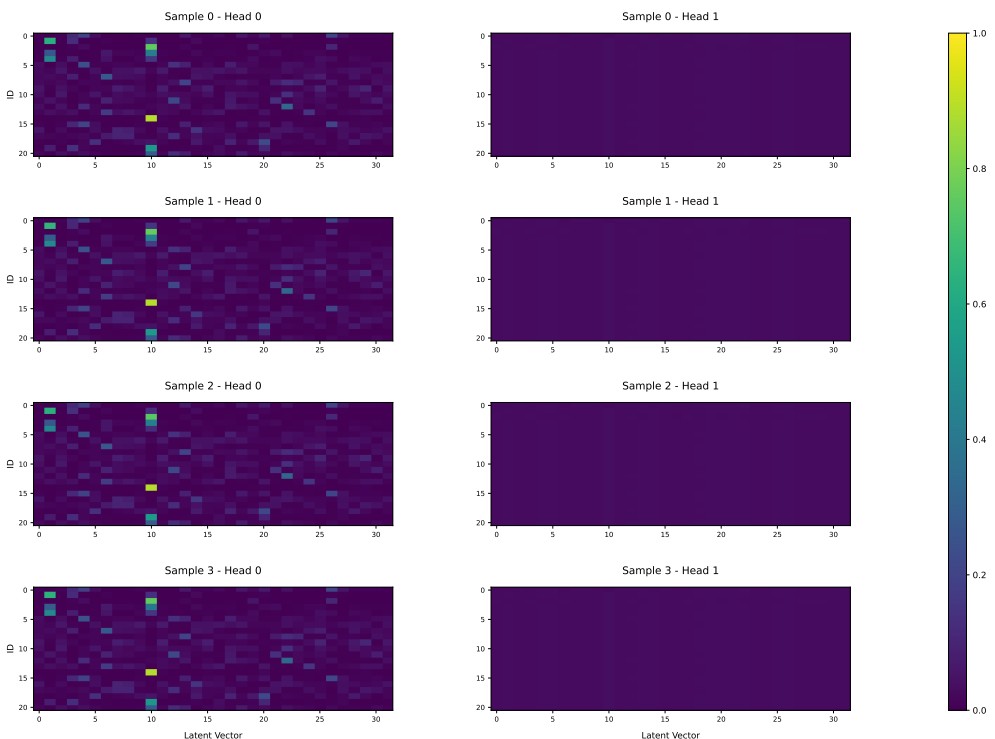

Figure 12: Attention patterns on the MD17 dataset for four random samples of the validation dataset, using a model trained with **16 latent vectors**. Each row corresponds to a single sample and displays the attention patterns across individual attention heads. Each sample employs the **same identifier assignment**.

# H Visualizations

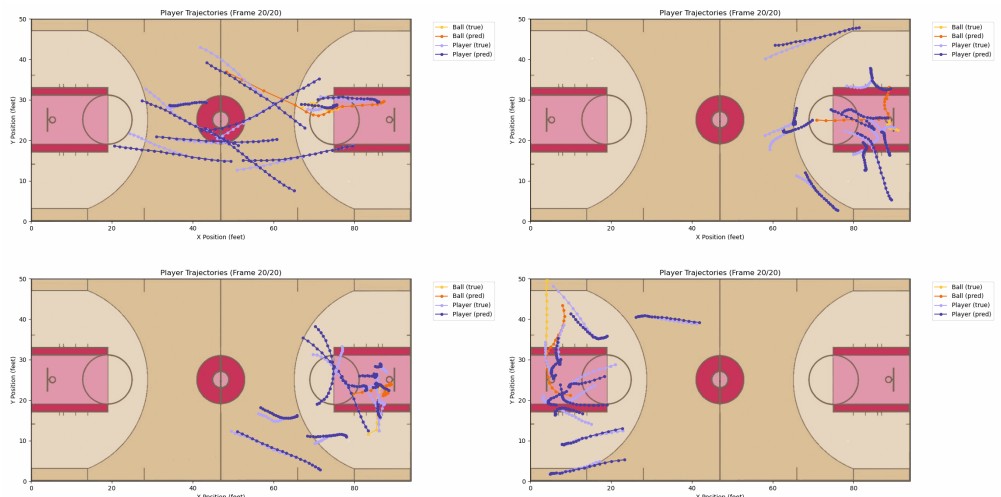

Figure 13: Basketball player trajectories from the NBA dataset. We visualize each player's movement path up to the final frame and include the ball's trajectory in the visualization. **Top row**: Trajectories for scoring scenarios. **Bottom row**: Trajectories for rebound scenarios. (Court image source: https://github.com/linouk23/NBA-Player-Movements/blob/master/court.png)

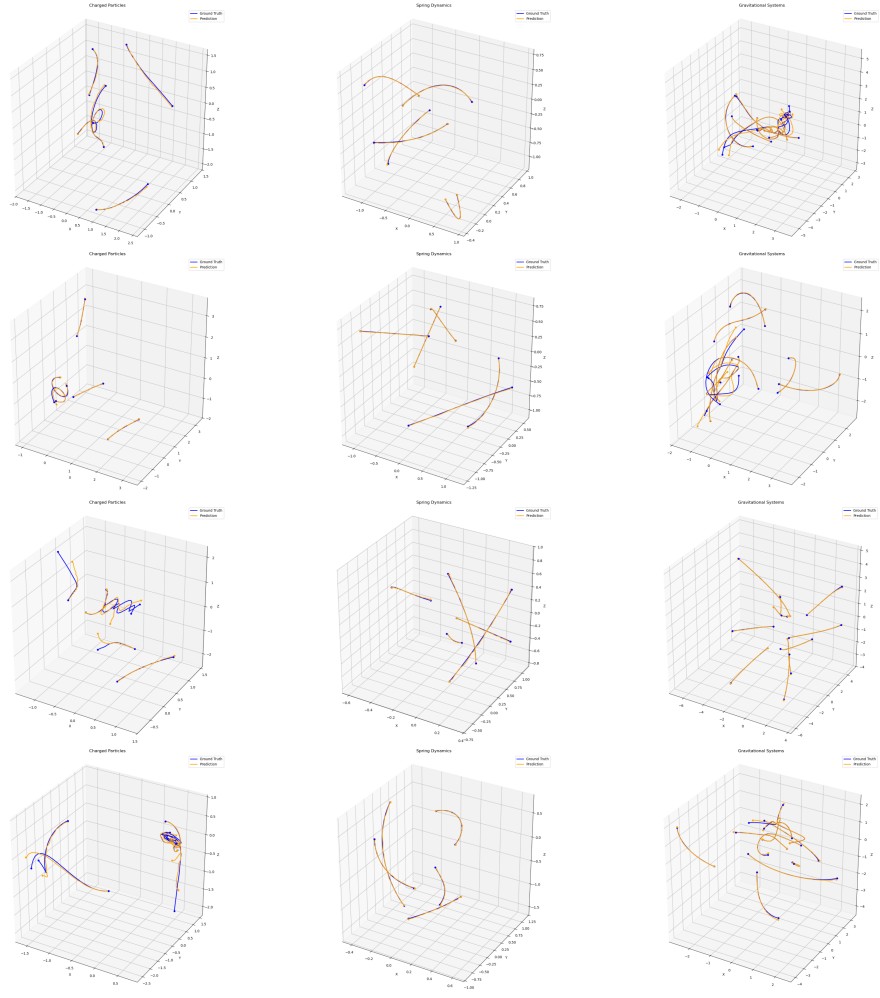

Figure 14: Trajectories from the N-Body dataset, predicted vs ground truth trajectories. **Left**: Charged particles. **Middle**: Spring dynamics. **Right**: Gravitational system.

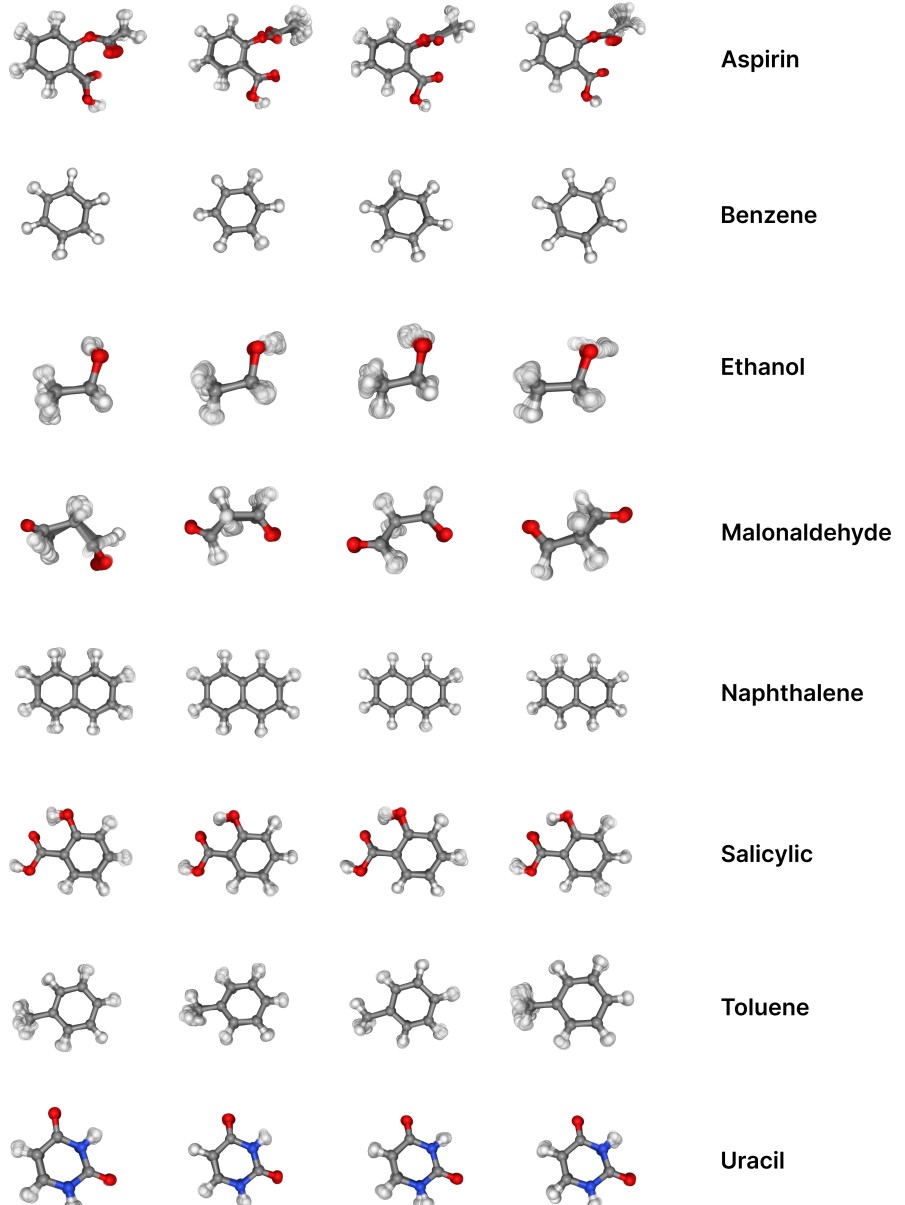

Figure 15: **Molecular dynamics trajectories from the MD17 dataset**, showing time-evolved structural predictions for each molecule. For every compound, we display four distinct trajectory predictions, with each prediction comprising 20 superimposed time frames to illustrate the range of conformational changes.

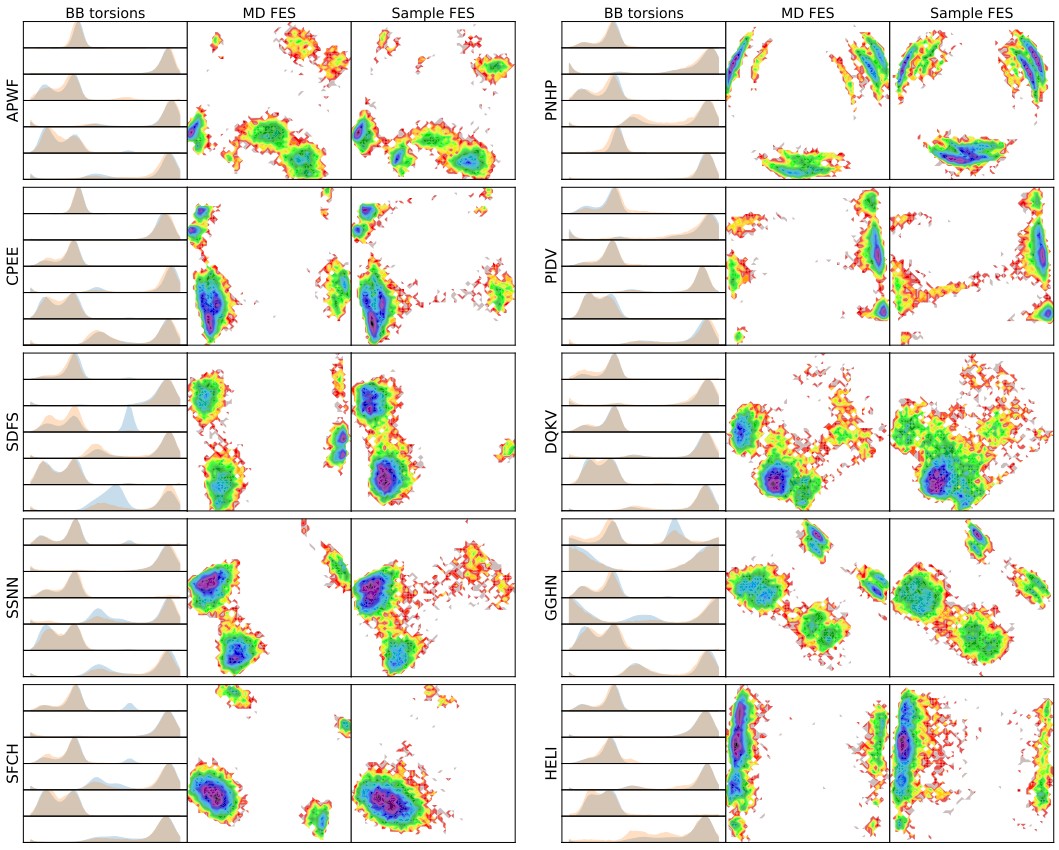

Figure 16: **Torsion angle distributions** of the six backbone torsion angles, comparing molecular dynamics (MD) trajectories (orange) and sampled trajectories (blue); and **Free energy surfaces** projected onto the top two time-lagged independent component analysis (TICA) components, computed from both backbone and sidechain torsion angles.

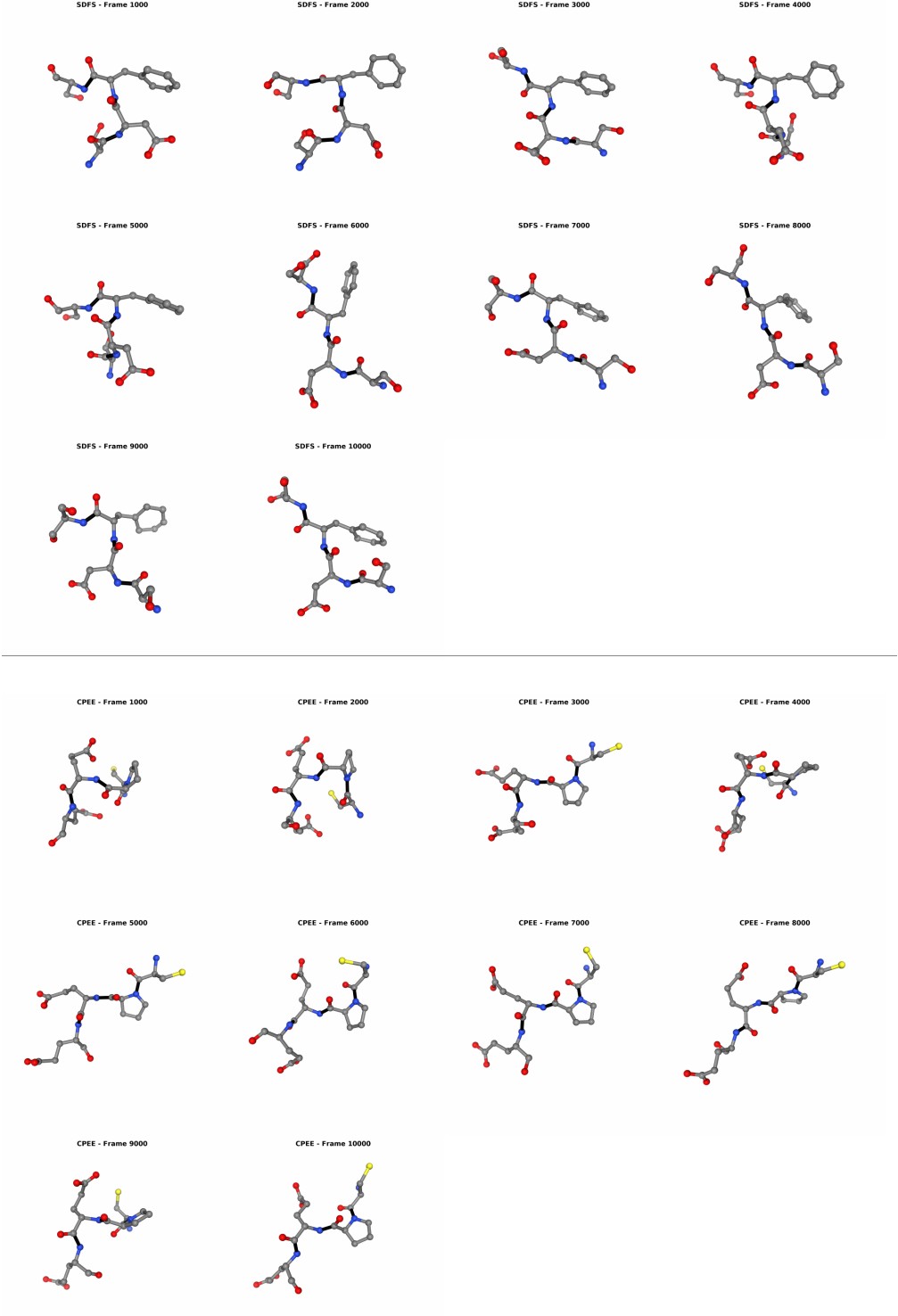

Figure 17: **Molecular Dynamics trajectories from the Tetrapeptides (4AA) dataset**, showing time-evolved structural predictions for ten frames at an interval of 1000 frames. **Top**: SDFS (Serine - Aspartic Acid - Phenylalanine - Serine) peptide. **Bottom**: CPEE peptide (Cysteine - Proline - Glutamic Acid - Glutamic Acid).

