# OpenReview forum: "LaM-SLidE: Latent Space Modeling of Spatial Dynamical Systems via Linked Entities"
_NeurIPS.cc/2025/Conference — NeurIPS 2025 poster_

### Official Review · Reviewer_8R2U · 2025-06-28

**Clarity:** 2
**Significance:** 2
**Originality:** 1
**Rating:** 4
**Confidence:** 2

**Summary:**

The goal of the paper is to bring the advantages of latent modeling from image generation models into dynamical systems models. One part of the method focuses on modeling the IDs of trajectories so that idintifiable reconstruction is possible after latent space.

**Questions:**

**Question 1**
>In order to leverage a latent systems representation, we need to be able to trace individual entities of the system

why? can you elaborate further. why needs to differ from image generators?
> my interpretation: because you are tracing elements, trying to train a generative model in latent space.

**Question 2**
Since this work is in latent space for dynamical system, how does it compare with methods such as: "Latent ODEs for Irregularly-Sampled Time Series" (NeurIPS 2019)? and why that type of work do not require IDs?

**Question 3**
> aspirin consists of 21 atoms, needs to have at least 21 unique identifiers

why? elaborate

**Ethical Concerns:**

["NO or VERY MINOR ethics concerns only"]

**Final Justification:**

The authors clarified that the method requires much less compute compared to GeoTDM. However, in my opinion, they should still discuss, and to what is possible, compare with, the related works that achieve better results. I previously put the link to 10 methods getting better performance.

**Limitations:**

yes

**Quality:**

2

**Strengths And Weaknesses:**

**Weaknesses**

W1: GeoTDM comparison. While GeoTDM is basically the most important baseline and all experiments are taken from there, the differences with this method are not discussed in any section.

W2: Missing baselines. At this link https://paperswithcode.com/sota/trajectory-prediction-on-ethucy the first top 10 methods have better performance than the method proposed by this paper for the ETH-UCY dataset. The task is from Table 2 from the submitted paper.

W3: the improvement over GeoTDM is marginal.

---

> ### Author Rebuttal · Authors · 2025-07-31
>
> Thank you for reviewing our submission. While your assessment was critical, we appreciate your suggestion to elaborate on the relationship to image generation, specifically why traceability does not appear to be an issue in this context. Specifically, our work presents novel ID representation that are where not applied in dynamical system modeling before, offering an alternative to GNN based approaches and leveraging recent progress in latent space image/ video modeling. We are looking forward to the discussion period and hope we can address open concerns.
>
>
> ### Weaknesses
>
> > GeoTDM comparison. While GeoTDM is basically the most important baseline and all experiments are taken from there, the differences with this method are not discussed in any section.
>
>
> While GeoTDM serves as an important baseline (in 3/5 experiments), we also want to highlight some conceptual differences between LaM-SLidE and GeoTDM. In particular, LaM-SLidE introduces a novel ID-based entity tracking mechanism which allows tracking entities in a fixed size latent space. To enable this, we relax certain equivariance properties [3] for our models — unlike GeoTDM. This design choice aligns with recent architectural trends, such as AlphaFold 3 [2], where similar relaxations have been argued to contribute to performance improvements. In response to this review we added a more detailed discussion in the results section. We also want to underline an advantage of GeoTDM: as shown in Table 15, LaM-SLidE requires significantly fewer function evaluations.
>
> Number of function evaluations (NFEs):
> |          | N-Body | MD17 | ETH |
> |----------|--------|------|-----|
> | GeoTDM | 1000   | 1000 | 100 |
> | LAM-SLIDE    | **10**     | **10**   |**10**  |
>
> > Missing baselines. At this link ... the first top 10 methods have better performance than the method proposed by this paper for the ETH-UCY dataset. The task is from Table 2 from the submitted paper.
>
> We appreciate your comment regarding the performance of recent methods on the ETH-UCY dataset. It is indeed true that several methods listed on the leaderboard achieve higher accuracy. However, we would like to mention in order to have a valuable baseline we relied on the evaluation pipeline introduced in GeoTDM, which could be different from the one used on this leaderboard. Further many of these approaches are highly specialized for human trajectory prediction, incorporating domain-specific inductive biases and handcrafted features. For example, the current top-performing method on the ETH-UCY dataset [1] models repulsion and attraction forces derived from the environment, using semantic segmentation to distinguish walkable areas, learnable goal positions, and other features specifically designed for human motion forecasting. We want to note that, in general, our model could also process additional node features, but we aimed for a proof-of-concept using the simplest version of the dataset to validate whether the ID-based representations can be leveraged across different domains.
>
>
> > The improvement over GeoTDM is marginal
>
> **Performance Comparison with GeoTDM**
> LaM-SLidE demonstrates clear performance advantages across multiple benchmarks. On the MD17 benchmark, LaM-SLidE achieves superior results compared to GeoTDM (Table 4). Similarly, in N-body system dynamics tasks, LaM-SLidE consistently outperforms GeoTDM, with the exception of scenarios where all methods achieve comparably low error rates.
> The ETH-UCY dataset presents more nuanced results, with less definitive performance differences between GeoTDM and LaM-SLidE. Given that pedestrian forecasting is inherently subject to behavioral stochasticity, the observed marginal performance variations may be attributable to random factors rather than fundamental algorithmic differences. Notably, LaM-SLidE offers significant computational speedup compared to GeoTDM.
>
> **Comparison with MDGen**
> We also encourage you to consider our comparison against MDGen shown below (Table 5). MDGen represents a highly specialized approach and an equivariant architecture operating in torsion space specifically designed for peptide molecules. Despite MDGen's domain-specific optimizations and restricted applicability to other domains, our general-purpose method achieves comparable performance, demonstrating the effectiveness of our approach across diverse domains.
>
> |           | Torsions |      |     | TICA |        | MSM  | Time   |
> |-----------|----------|------|-----|------|--------|------|--------|
> |           | BB       | SC   | All | 0    | 0.1 joint |      |        |
> | 100ns     | .103     | .055 | .076| .201 | .268   | .208 | ~ 3h   |
> | MDGen     | .130     | **.093** | **.109**| .230 | .316   | .235 | ~ 60s  |
> | LAM-SLIDE | **.128**     | .122 | .125| **.227** | **.315**   | **.224** | ~ 53s  |
>
> LaM-SLidE achieves competitive performance across most metrics while maintaining competitive computational efficiency to MDGen (53s vs 60s). This also highlights the strong potential of deep learning–based surrogate models in the simulation domain.
>
> > In order to leverage a latent systems representation, we need to be able to trace individual entities of the system. Why? can you elaborate further. why needs to differ from image generators? my interpretation: because you are tracing elements, trying to train a generative model in latent space.
>
> Image generators compute pixel values for each location within a structured grid, where every pixel is inherently tied to a fixed coordinate. In such representations, pixels act as entities whose positions are implicitly defined by their location in the grid. The pixel values are typically derived from a latent image representation, and the underlying neural network architecture dictates how each specific grid position (i.e., pixel value) is computed from the latent space. The relationship between latent features and pixels is fixed by design.
>
> In contrast, the dynamical systems considered here, represent states using a set of discrete entities that are not bound to static spatial positions, and the number of entities is not fixed. These entities are free to move, and their positions can evolve over time. Without identifiers, the system would compress all entities into a shared latent representation that no longer has links to individual entities. Because of this loss of direct association, unique identifiers are required to maintain reference to each entity.
> This identity-based, set-oriented approach is especially effective when entities are sparsely distributed across space. In such cases, tracking entities by their identifiers is significantly more efficient than scanning a dense grid volume to detect their presence.
>
>
> **References**
>
> [1] J. Yue et al., Human Trajectory Prediction via Neural Social Physics.
> [2] J. Abramson et al.,  Accurate structure prediction of biomolecular interactions with AlphaFold 3.
> [3] J. Brehmer et al., Does equivariance matter at scale?

---

> > ### Comment · Reviewer_8R2U · 2025-08-02
> >
> > >It is indeed true that several methods listed on the leaderboard achieve higher accuracy.
> >
> > In my opinion, some of those 10 methods should be considered related works, discussed, and possibly, compared with. You only mentioned about 1/10. If you agree, making that effort to discuss those methods could be useful.
> >
> > Importantly, since your method also achieves a speedup compared to GeoTDM, I decide to raise my score.

---

> > > ### Author Response · Authors · 2025-08-04
> > >
> > > Thank you for your constructive feedback and for recognizing the value of our method's computational efficiency. We appreciate your decision to raise your score, and agree that a more comprehensive discussion of these methods strengthens our paper. We are adding those.

---

### Official Review · Reviewer_4WJU · 2025-07-01

**Clarity:** 3
**Significance:** 4
**Originality:** 4
**Rating:** 5
**Confidence:** 4

**Summary:**

This paper proposes LaM-SLidE, a novel framework for modeling spatial dynamical systems using latent space generative modeling with traceable individual entities. The method combines encoder–approximator–decoder architecture with identifier-based latent representations, allowing traceability of individual particles even when modeling is performed in a compressed latent space. A core feature is the ability to train and generalize across systems with variable numbers of entities, enabling parameter sharing and representation unification. The authors evaluate the method on a range of tasks, including pedestrian dynamics, basketball plays, molecular dynamics, and N-body systems, showing competitive or superior performance against a broad set of baselines.

**Questions:**

1. Lines 104–106 introduce $M^t$, but immediately state it is time-independent. Can this symbol be omitted in favor of a single static matrix $M$? Similarly, equations (1) and (2) formalize the Markov property, but this property is not used anywhere in the model. Would the paper be clearer without these?

2. The paper defines the stochastic interpolation path in Eq. (6), with boundary conditions $\alpha_1 = \sigma_0 = 0$, $\alpha_0 = \sigma_1 = 1$. Shouldn't this be the opposite? Please clarify.

3. The description of the approximator starting at line 180 is difficult to follow. Even after consulting Appendix B.4 and C, the architectural details and training procedure are unclear. Could the authors provide a more complete and explicit description of the model, perhaps with a step-by-step example or pseudocode?

4. How does the model adapt to variable-length conditioning ($T_o$)? The framework appears to handle different observation lengths, but it is not clear how the approximator handles this in training or inference. Is $T_o$ used as an explicit condition, or is the model trained over varying $T_o$ lengths? This point deserves clarification.

5. Why not use a step-wise (Markov) prediction? Given that the dynamics are assumed to be Markovian (as implied by Equations 1 and 2), why is the model trained to predict the entire future trajectory at once, rather than iteratively predicting the next step? A step-wise approach may allow for simpler and more interpretable modeling.

6. For systems of identical particles (e.g., molecules with symmetric atoms), how does LaM-SLidE handle permutation invariance? The current identifier mechanism seems to break symmetry. Could the authors discuss the implications and whether permutation equivariance could be incorporated into this framework?

**Ethical Concerns:**

["NO or VERY MINOR ethics concerns only"]

**Final Justification:**

Thank authors for their responses and discussions. I believe they have addressed my concerns very well, and I support the acceptance of this paper to the conference.

**Limitations:**

Yes

**Paper Formatting Concerns:**

No major formatting issues were found.

**Quality:**

4

**Strengths And Weaknesses:**

Strengths:

1. The paper tackles an important and nontrivial challenge—how to apply latent generative models to physical systems with variable-sized, traceable entities. The proposed approach is both technically novel and practically impactful.
2. One of the most compelling aspects is the ability to jointly model systems of varying particle counts in a shared latent space, a feature often lacking in prior GNN- or diffusion-based methods.
3. Experiments are comprehensive and cover diverse domains.
4. The proposed framework is modular and general enough to be applied across domains.

Weaknesses:

1. Some symbols are defined but immediately undermined, or unused in the actual modeling (e.g., $M^t$ and equations (1–2)), leading to unnecessary conceptual load.
2. The description of the latent dynamics approximator (especially from line 180 onwards) is vague and scattered. Even after reading Appendix B.4 and C, it remains unclear how the approximator operates in detail.
3. While the model appears to support different input lengths $T_o$, it is unclear how this is explicitly handled.
4. The paper does not explain why the modeling is not done step-wise in time if the system is Markovian. Such an autoregressive approach might be simpler and more interpretable.
5. The paper lacks discussion on how the model handles or violates permutation invariance among identical particles, a property crucial in many physical systems.

---

> ### Author Rebuttal · Authors · 2025-07-31
>
> We appreciate that you acknowledged the technical novelty and practical impact of our approach, particularly highlighting our ability to jointly model systems with varying particle counts in a shared latent space, a capability you noted is often missing in prior methods. We also appreciate your recognition of our comprehensive experimental validation and the framework’s general applicability. We hope our responses below address the concerns you raised about our work.
>
> ### Weaknesses / Questions
>
> > Lines 104–106 introduce , but immediately state it is time-independent. Can this symbol be omitted in favor of a single static matrix ? Similarly, equations (1) and (2) ...
>
>
> Regarding $M^t$: While we could simplify to a static matrix M, we chose to retain the time-dependent notation $M^t$ to emphasize the generality of our framework. This notation makes explicit that our approach could naturally extend to systems with time-varying entitiy properties except positions, this aligns with the flexibility discussed in response to question 2 of reviewer PyHh.
>
> Regarding Equations (1) and (2): Our aim was to provide readers, especially those unfamiliar with dynamical systems, with a conceptual introduction to the topic. While we understand the your concern, we respectfully disagree in this particular point with the assertion that the Markov property is entirely absent. As the your correctly noted elsewhere, our approach does not involve single-step trajectory predictions. Instead, we employ a block-wise rollout strategy, where entire trajectory segments are predicted step-by-step. This process can be interpreted as Markovian at the level of each block. Predicting multiple time steps also offers computational advantages, which we discuss below.
>
>
> > The paper defines the stochastic interpolation path in Eq. (6), with boundary conditions , . Shouldn't this be the opposite? Please clarify.
>
> You are correct, this was a typo. The correct boundary conditions should be $\alpha_0=\sigma_1=0$ and $\alpha_1=\sigma_0=1$. We corrected it in the revised manuscript.
>
> > The description of the approximator starting at line 180 is difficult to follow. Even after consulting Appendix B.4 and C, the architectural details and training procedure are unclear. Could the authors provide a more complete and explicit description of the model, perhaps with a step-by-step example or pseudocode?
>
> We apologize for any confusion and have included Python pseudocode below for better clarity. To clarify our contributions: while our latent approximator uses alternating attention blocks across temporal and spatial dimensions, a commonly used approach in latent video modeling, our primary contribution is the flexible fixed-size latent space that allows us to effectively utilize these architectural components.
>
>
> ```python
> def latent_layer(z, t, z_cond):
>     ## Latent layer of the approximator (high level)
>     z = z + embed_cond(z_cond)
>     t_embed = embed_time(t)
>
>     z = attention_spatial(z, t_embed)     # T x L x Dz, attention is calculated over L    | ref [4,5]
>     z = rearrange(z, "T L Dz -> L T Dz")
>     z = attention_temporal(z, t_embed)    # L x T x Dz, attention is calculated over T    | ref [4,5]
>     z = rearrange(z, "L T Dz -> T L Dz")
>
>     return z
>
>
> ## Training (Python Pseudocode)
> ## Assume a trained encoder/decoder
> ## For easier illustration we omit the batch dimension and show the training procedure for a single samplel
>
> ##################
> x                                     # T x N x Dn (N=number of entities, D=[x,y,z,atom_type]) ... e.g. Aspirin 21x4
> ids                                   # N e.g. Aspirin 21
> mask                                  # T binary mask [1, 0, 0, ....], e.g. one conditioning frame
> ##################
>
> # Encoding and conditioning setup
> z1 = encode(x, ids)                   # T x L x Dz Encoder state into latent vector
> z_cond = mask_frames(z1, mask)        # T x L x Dz Contains only the conditioning frames, the other frames are masked out and set to 0.
>
> # SiT path
> z0 = torch.randn_like(z1)             # Sample z0 from a normal distribtion
> t = torch.uniform(0, 1)               # Sample t uniformly
> zt = t * z1 + (1 - t) * z0            # Create interpolation
>
> # Loss
> z1_hat = approximator(zt, t, z_cond)
> loss = torch.mean((z1_hat - z1)**2)
> loss.backward()
>
>
> ## Inference
> # We use the identiy from ref [11] Table 1,
> # to transform a data prediction model into a velocity model, and use
> # torchdiffeq (https://github.com/rtqichen/torchdiffeq) for integration
>
> ##################
> x_cond                                                   # T x N x Dn, but non-conditioning frames are zero
> ##################
>
> velocity_model = data_to_velocity_model(approximator)    # Transform model trained on data prediction into velocity prediction model.
>
> z_cond = encode(x_cond, ids)                             # T x L x Dz, contains only the conditioning frames/ missing frames are set to 0
> z0 = torch.randn_like(z1)
> z1_pred = solver(velocity_model, z0, z_cond)             # Solve ODE
>
> x_hat = decoder(z1_pred)
> ```
>
> We've also recognized, that some figures may not render properly on Mac systems, particularly Figure 5 which depicts the overall flow of our approach. These figures display correctly when viewed in Firefox browser on Mac systems.
>
> > How does the model adapt to variable-length conditioning ($T_0$)? The framework appears to handle different observation lengths, but it is not clear how the approximator handles this in training or inference. Is it used as an explicit condition, or is the model trained over varying  lengths? This point deserves clarification.
>
> You are correct, that we use explicit conditioning with fixed lengths for each task during training and inference. So we handle different observation lengths, but explicitly trained for each task. Our current experiments use the following number of conditioning frames (see also Table 7 in our manuscript):
>
> | Experiment | Conditioning Frames |
> |------------|-------------------|
> | Pedestrian trajectory forecasting (ETH-UCY) | 8 |
> | Basketball (NBA) | 8 |
> | N-Body | 10 |
> | Molecular Dynamics (MD17) | 10 |
> | Molecular Dynamics - Tetrapeptides (4AA) | 1 |
>
> We acknowledge that variable-length conditioning represents a promising extension. Recent video modeling work demonstrates the benefits of this approach: [7] trains with varying conditioning lengths with different noise levels, while [10] employs random frame masking for video modeling. Futher, [9] showed strong results modeling PDEs, while randomly masking conditioning frames during training. While we have not yet explored variable-length conditioning, we think it could enhance both performance and flexibility.
>
>
> > Why not use a step-wise (Markov) prediction? Given that the dynamics are assumed to be Markovian (as implied by Equations 1 and 2), why is the model trained to predict the entire ...
>
> While the underlying dynamics are Markovian (as reflected in Equations 1 and 2), we opt for non-autoregressive, full-trajectory prediction following recent advances in video generation and simulation modeling [6,7]. These works have demonstrated that predicting multiple future steps jointly can lead to improved performance in terms of both accuracy and stability, especially for long-range forecasting tasks. Recently [9] used a mutli-step  prediction approach for PDEs, showing remarkable results.
>
> From a practical perspective, full-trajectory prediction also brings substantial computational benefits. For instance, in the Tetrapeptides dataset, we predict 1000 future timesteps in parallel, resulting in a significant speedup compared to autoregressive rollouts, which require sequential evaluation of each step.
>
> > For systems of identical particles (e.g., molecules with symmetric atoms), how does LaM-SLidE handle permutation invariance? The current identifier mechanism seems to break symmetry. Could the authors discuss the implications and whether permutation equivariance could be incorporated into this framework?
>
> In molecular simulations, each atom is individually tracked throughout the simulation, maintaining consistent identity across the entire trajectory. This means that even for molecules with permutation symmetries, the identity of each atom remains fixed, effectively sidestepping the permutation equivariance/invariance issue. While we acknowledge that permutation equivariance/invariance is an important property in certain contexts [6,8], we have opted to omit it in favor of simplicity and computational efficiency, following also recent trends [1,2,3] not explicitly incorporating these inductive biases into our architecture. We discuss our data augmentation approach in Appendix Section E.3, which partially addresses symmetry considerations. We think, that incorporating permutation equivariance/invariance or other inductive biases remains an interesting direction for future work and added a discussion on it into our manuscript.
>
> **References:**
>
> [1] J. Brehmer et al., Does equivariance matter at scale?
> [2] K. Joshi et al., All-atom Diffusion Transformers: Unified generative modelling of molecules and materials
> [3] J. Abramson et al.,  Accurate structure prediction of biomolecular interactions with AlphaFold 3
> [4] W. Peebles et al., Scalable Diffusion Models with Transformers
> [5] M. Dehghani et al., Scaling Vision Transformers to 22 Billion Parameters
> [6] B. Jing et al., Generative Modeling of Molecular Dynamics Trajectories
> [7] B. Chen et al., Diffusion Forcing: Next-token Prediction Meets Full-Sequence Diffusion
> [8] D. Knigge et al., Space-Time Continuous PDE Forecasting using Equivariant Neural Fields
> [9] F. Rozet et al., Lost in Latent Space: An Empirical Study of Latent Diffusion Models for Physics Emulation
> [10] V. Voleti et al., MCVD: Masked Conditional Video Diffusion for Prediction, Generation, and Interpolation
> [11] Y. Lipman et al., Flow Matching Guide and Code.

---

> > ### Comment · Reviewer_4WJU · 2025-08-06
> >
> > Thank you for the authors’ responses. I believe they have addressed my concerns very well, and I support the acceptance of this paper to the conference. My only remaining suggestion is whether the authors could consider adding a pseudocode summary of the proposed model and algorithm at an appropriate place in the manuscript. This would help readers better grasp the overall structure of the method and offer a more readable alternative to the actual Python implementation.

---

> > > ### Author Response · Authors · 2025-08-06
> > >
> > > Thank you for supporting our paper's acceptance and confirming our response addressed your open concerns. Following your suggestion, we will include a complete pseudocode for training and inference.

---

### Official Review · Reviewer_PyHh · 2025-07-03

**Clarity:** 3
**Significance:** 2
**Originality:** 3
**Rating:** 4
**Confidence:** 3

**Summary:**

This paper introduces a novel approach that replaces GNNs for modeling spatial dynamical systems, using modern latent space generative modeling techniques instead. The key innovation is an identifier (ID) system where each entity is assigned a unique ID with learnable embeddings. These IDs are concatenated with entity positions and features, then encoded via cross-attention to a fixed number of latent tokens. A flow-matching model operates in this latent space for temporal dynamics, and a decoder uses ID-based queries to retrieve entity-specific information. The method is evaluated across diverse domains including molecular dynamics (MD17, Tetrapeptides), human motion (pedestrian, basketball), and N-body systems, achieving state-of-the-art or competitive performance while requiring 10-100x fewer function evaluations than baseline methods.

**Questions:**

Please see the weakness part above,

**Ethical Concerns:**

["NO or VERY MINOR ethics concerns only"]

**Final Justification:**

The rebuttal has addressed all my concerns. Based on the effectiveness of this method on modeling dynamics, and the overall positive rating of this paper, I would like keep my rating as WA.

**Limitations:**

Please see the weakness part above,

**Quality:**

3

**Strengths And Weaknesses:**

======

Strengths:

*Eliminates graph construction overhead: No need to learn to cluster and build the graph, as in GNN; learns to encode directly into fixed-size latent space, avoiding k-NN computations and edge message passing.

*Enables modern architectures: The fixed-size latent representation enables usage of SOTA scalable generative models (flow matching, diffusion models) that have driven recent advances in image/video generation.

*Superior performance: Achieves better performance than GNN-based methods across all tested domains, notably beating equivariant GNNs on MD17 without using domain-specific symmetries.

*Unified architecture: Single model can handle molecules of different sizes (9-21 atoms in MD17), demonstrating better generalization than molecule-specific training.

=====

Weakness

*Hyperparameter sensitivity: As mentioned in the paper, the pool size for potential entities matters. But tweaking this hyperparameter requires trial-and-error or domain expertise. More importantly, is it necessary to retrain the model for different pool sizes? Can models learned with different pool sizes share common knowledge such that we don't have to learn from scratch?


*Dynamic entity handling is unclear: How does the method handle changes in entity count during dynamics? What if there are new entities entering the system? How are entity mergers or departures handled?


*Tracking dependency: The method relies on perfect temporal tracking being provided by the dataset. How sensitive is the approach to tracking errors? What happens with identity switches or occlusions?

*Semantic analysis missing: The paper needs more discussion on the semantic meaning of assigned entities given N entities and L latent vectors:
For N<L (overparameterization), do multiple latents correspond to the same entity or capture different aspects?
For N≥L (compression), does this lead to semantically meaningful clustering in latent space?
I appreciate the brief discussion and results in Appendix G.3 on number of latent vectors, G.4 on pool size effects , though more extensive ablations would indeed clarify this.

*(minor)Incomplete ablations: What drives the performance gains - the ID system, the flow matching model, or simply the larger model capacity? More ablations comparing to fixed-size GNN baselines would strengthen claims.


In summary,
this paper presents an innovative approach to modeling spatial dynamical systems by replacing the dominant GNN paradigm with fixed-size latent representations. The core contribution - the ID-based encoding/decoding system - elegantly solves the problem of maintaining entity identity while leveraging modern generative architectures. The experimental results are impressive, showing consistent improvements across diverse domains. So I'm leaning towards the acceptance of the paper.

Having said that,
 several limitations temper my enthusiasm. The requirement to pre-specify maximum entity counts and retrain for different pool sizes significantly limits the "foundation model" potential. The dependence on perfect tracking and inability to handle dynamic entity counts restrict real-world applicability. The paper would benefit from deeper analysis of what the model learns (e.g., embedding structure) and more thorough ablations to isolate the sources of improvement.

---

> ### Author Rebuttal · Authors · 2025-07-31
>
> We appreciate your comprehensive assessment and recognition of our key contributions: eliminating graph construction overhead, leveraging modern scaleable architectures, and achieving strong performance without domain-specific inductive biases. We are grateful for the opportunity to strengthen our work through addressing the concerns raised. Your suggestion for semantic analysis is particularly valuable, leading to new insights about our model's internal representations that we had not previously explored.
> We also agree, that "foundational architecture" may overstate our current scope and we adjusted our wording accordingly. Our intention was to highlight the flexibility of ID-based approaches across different dynamical systems. While limitations like pre-specified entity counts exist, we believe this work contributes a novel paradigm that opens promising research directions.
>
>
>
> ### Weaknesses
>
> > Hyperparameter sensitivity: As mentioned in the paper, the pool size for potential entities matters. But tweaking this hyperparameter requires trial-and-error or domain expertise. More importantly, is it necessary to retrain the model for different pool sizes? Can models learned with different pool sizes share common knowledge such that we don't have to learn from scratch?
>
> We agree pool size represents an important hyperparameter requiring consideration. However, Figure 7 indicates this limitation is less restrictive than initially apparent, where reconstruction errors increase marginally when pool size exceeds natural dataset requirements (MD17's 21 entities). The performance degradation appears to result from computational constraints (reduced embedding update frequency) rather than model capacity limitations, suggesting that extended training or transfer learning strategies could enable effective knowledge sharing across different pool configurations without requiring complete retraining. To summarize, we can safely use too many IDs, it just takes longer to train.
>
> > Dynamic entity handling is unclear: How does the method handle changes in entity count during dynamics? What if there are new entities entering the system? How are entity mergers or departures handled?
>
> While the scope of current experiments did not explicitly cover scenarios with changing entity counts or new entities entering the system, we think our architecture is flexible enough to handle these cases.
>
> Currently, entity attributes (such as atom type in molecular dynamics or team membership in basketball) remain static throughout the dynamics. However, our framework could incorporate such behavior by introducing time-varying properties such as "active/inactive" or "groups assignment" to the entities. This would enable modeling of:
>
>  - Entity entry and exit events
>  - Entity mergers and splits
>
> We acknowledge that this as a valuable direction for future research and added it to the discussion section of our manuscript.
>
>
> > Tracking dependency: The method relies on perfect temporal tracking being provided by the dataset. How sensitive is the approach to tracking errors? What happens with identity switches or occlusions?
>
> While we have not explicitly tested our method's sensitivity to tracking errors, we acknowledge this as a crucial consideration for real-world applications. Our current approach assumes perfect tracking, which is reasonable for simulation-based datasets (e.g., molecular dynamics). However, we recognize that real-world tracking systems are prone to sensor errors and occlusions. Future work could address this limitation by introducing synthetic tracking errors during training, such as randomly dropping entities to improve model robustness.
>
>
> > Semantic analysis missing: The paper needs more discussion on the semantic meaning of assigned entities given N entities and L latent vectors: For N<L (overparameterization), do multiple latents correspond to the same entity or capture different aspects? For N≥L (compression), does this lead to semantically meaningful clustering in latent space? I appreciate the brief discussion and results in Appendix G.3 on number of latent vectors, G.4 on pool size effects , though more extensive ablations would indeed clarify this.
>
> For a better understanding of the semantic structure of the latent space, we conducted additional experiments.
>
>  **Experimental Setup:**
> We analyzed decoder attention weights to understand entity-latent correspondence in both overparameterized (N<L) and compressed (N≥L) regimes. Using two first-stage models (L=16 and L=32) trained on MD17, we examined how they encode aspirin molecules (N=21 atoms) across different conformations. Due to NeurIPS rebuttal constraints, we present our findings textually here but included the visualizations in the revised manuscript.
>
>  **Key Findings:**
>  - **Consistent attention patterns:** When encoding different aspirin conformations with consistent atom IDs, we observe identical attention patterns per ID, which is distributed over the latent vectors. This suggests that each ID learns a unique "addressing scheme" for retrieving its information from the latent space (independent of the stored information).
>  - **Adaptive attention head utilization:** The models exhibit different strategies based on compression ratio:
>    - **L=32 (overparameterized):** Leverages only a single attention head, we think the ID based retrieval acts like a hashing table and there is no need for the model to utilize both heads if the latent space is large enough
>    - **L=16 (compressed):** In the compressed setting we saw unique attention patterns in both heads, suggesting the model needs to avoid potential collisions with other IDs.
>
>
>
> > Incomplete ablations: What drives the performance gains - the ID system, the flow matching model, or simply the larger model capacity? More ablations comparing to fixed-size GNN baselines would strengthen claims.
>
>
> Our approach builds on recent progress in latent-space image and video modeling [1,2,3], and we aimed to apply these concepts to dynamical systems with a varying number of entities, a setting currently dominated by GNNs. We would be happy to include specific comparisons to fixed-size GNNs in the manuscript if you have a particular suggestion.
>
> **References**
>
> [1] A. Polyak et al., Movie Gen: A Cast of Media Foundation Models
> [2] P. Esser et al., Scaling Rectified Flow Transformers for High-Resolution Image Synthesis
> [3] R. Rombach et al., High-Resolution Image Synthesis with Latent Diffusion Models

---

### Official Review · Reviewer_fy2s · 2025-07-03

**Clarity:** 3
**Significance:** 3
**Originality:** 3
**Rating:** 4
**Confidence:** 2

**Summary:**

This paper introduces a model for generative modeling of stochastic trajectories. The model bridges the traceability of individual entities and the latent system representation. It uses the identifiers (IDs) that allow for the retrieval of entity properties. The author conducted a comprehensive experiment to validate the model.

**Questions:**

1. Could the author provide more insights into the results on the ETH-UCY and the NBA dataset, where LAM-SLIDE only shows marginal or no improvement?
2. Does the author have any insights on whether the number of particles in the system affects the model performance?
3. Could the author provide some explanation on why LAM-SLIDE 2.1M seems to underperform GeoTDM 1.9M (all→each) in Table 13?

**Ethical Concerns:**

["NO or VERY MINOR ethics concerns only"]

**Final Justification:**

The author has addressed all my questions. The quality of the work meets the standard for a NeurIPS acceptance. However, I still believe further evaluation could be done on this work. Thus, I will retain my score as it is.

**Limitations:**

Yes.

**Quality:**

3

**Strengths And Weaknesses:**

**Strengths**
1. The model introduces a novel approach by using identifiers (IDs) to allow traceability, closing the gap in previous generative modeling methods.
2. The model has good performance across different domains, showing the general effectiveness of the approach.

**Weaknesses**
1. The paper offers only limited discussion and ablation studies. More ablations on the design choices, such as identifiers (IDs), will be beneficial.
2. The number of parameters for LAM-SLIDE is not indicated in Table 14.

---

> ### Author Rebuttal · Authors · 2025-07-30
>
> We greatly appreciate your constructive feedback, recognizing both our technical contributions (ID-based traceability and cross-domain performance) and opportunities to strengthen the presentation of our work. Below, the you can find the response to the raised concerns.
>
> ### Weaknesses
>
> > The paper offers only limited discussion and ablation studies. More ablations on the design choices, such as identifiers (IDs), will be beneficial.
>
> We agree that our ablation studies on identifiers (IDs) deserve more prominence in the main paper. Although we conducted extensive experiments (Appendix G), these findings were not effectively communicated in the main text. Our key results include the following, with the fourth point representing new analysis conducted for this rebuttal:
>
> - **ID pool size (Figure 7)**
> Reconstruction errors remain low (angstrom scale) across the ablated pool sizes. Since our ablation used the same number of training epochs across all pool sizes, larger pools naturally result in less frequent updates for the individual ID embeddings. So we can use much more possible IDs during training than entities, but because only a subset of the IDs are assigned in each update step, an increased pool size would benefit from longer training times.
>
> - **Latent vector count (Figure 8)**
> The reconstruction quality remains stable even under significant compression, where the number of latents $L$ is much smaller than the number of entities $N$. In the extreme case of $L=4$ performance degrades noticeably.
>
> - **Robustness of the ID assignment**
> The error bars in Figures 7 and 8 show the results of different random ID assignments, which produce nearly identical outcomes, even with large pool sizes (512 IDs). This indicates that the model learns ID-invariant representations rather than memorizing specific ID-to-entity mappings, a crucial property for generalization.
>
> - **Learned / Orthogonal ID embeddings**
> We compared fixed orthogonal ID embeddings as an alternative to learned embeddings. While orthogonal initialization also works, learned ID embeddings consistently deliver better results. Note that orthogonality is only approximate when the ID pool size exceeds the embedding dimension, as true orthogonality requires the number of IDs ≤ embedding dimension.  t-SNE visualization shows learned IDs distribute evenly across the embedding space.
>
> In response to your feedback, we moved these ablation studies from Appendix G to a new Section 4.7 ("Analysis of Design Choices") in the main paper. This section includes our existing analyses of identifier pool size and assignment strategies, augmented with an additional ablation conducted in the response of reviewer PyHh.
>
> > The number of parameters for LAM-SLIDE is not indicated in Table 14.
>
> You are right there there is an inconsistency in notation. While the parameter count appears in the second-to-last column, we agree it should be positioned next to the method name as in the other tables. We updated Table 14 in our manuscript and provide a markdown version of the updated table below. The 100ns method is a reference simulation acting as lower bound and does not have parameters.
>
>  | Method | Torsions |  |  | TICA |  | MSM | Time |
>  |--------|---------|---------|---------|---------|---------|---------|---------|
>  |  | BB | SC | All | 0 | 0,1 joint |  | (M) |
>  | 100 ns | .103 | .055 | .076 | .201 | .268 | .208 | ~ 3h |
>  | MDGen 34M | .130 | **.093** | **.109** | .230 | .316 | .235 | ~ 60s |
>  | LAM-SLIDE 28M | **.128** | 0.122 | 0.125 | **.227** | **.315** | **.224** | ~ 53s |
>  | LAM-SLIDE 11M | .152 | .151 | .152 | .239 | .331 | .226 |  |
>  | LAM-SLIDE 7M | .183 | .191 | .187 | .26 | .356 | .235 |  |
>  | LAM-SLIDE 4M | .284 | .331 | .311 | .339 | .461 | .237 |  |
>
> ### Questions
>
> > Q1: Could the author provide more insights into the results on the ETH-UCY and the NBA dataset, where LAM-SLIDE only shows marginal or no improvement?
>
> We agree, that the improvements on the ETH-UCY dataset are not as strong as on the other datasets. However, we would like to note that the baseline methods are specifically designed for this type of task and often rely on prior knowledge about relationships between entities, such as inter-entity distances. Additionally, they additionally incorporate extra features like velocity and acceleration. In contrast, our method does not make such assumptions or require these additional features. While our model could also benefit from task-specific feature engineering, we leave such ablations to future work.
>
> > Q2: Does the author have any insights on whether the number of particles in the system affects the model performance?
>
> We investigated this using the MD17 dataset and found no clear correlation between particle count and model performance. Instead, molecular flexibility appears to be the dominant factor here. As an intuitive example, despite having fewer atoms, Ethanol (9 atoms) shows higher prediction error than Naphthalene (18 atoms) due to its greater conformational flexibility. This pattern is evident in Table 4 and visualized in Figure 11, where Ethanol's dynamic hydroxyl group creates more complex trajectories than Naphthalene's rigid aromatic structure, suggesting system complexity, rather than raw particle count, drives prediction difficulty.
>
> > Q3: Could the author provide some explanation on why LAM-SLIDE 2.1M seems to underperform GeoTDM 1.9M (all→each) in Table 13?
>
> We would appreciate clarification on this observation, as we do not immediately see where LAM-SLIDE 2.1M underperforms GeoTDM 1.9M (all→each) in Table 13. We believe a potential misinterpretation could stem from our underlining approach. In Table 13 we underlined the second-best non-LAM-SLIDE method rather than the overall second-best method, which we recognize could lead to confusion. To address this, we have updated the table in our manuscript by adding two additional columns showing the ADE/FDE mean for each row, and present the last five rows of the updated below. We hope these changes make the results more accessible for the reader.
>
> |                          | Aspirin |        | Benzene |        | Ethanol |        | Malonaldehyde |        | Naphthalene |        | Salicylic |        | Toluene |        | Uracil |        | Mean     | Mean     |
> |--------------------------|---------|--------|---------|--------|---------|--------|---------------|--------|-------------|--------|-----------|--------|---------|--------|--------|--------|----------|----------|
> |                          | ADE     | FDE    | ADE     | FDE    | ADE     | FDE    | ADE           | FDE    | ADE         | FDE    | ADE       | FDE    | ADE     | FDE    | ADE    | FDE    | ADE      | FDE      |
> | GeoTDM 1.9M              | 0.107   | 0.193  | 0.023   | 0.039  | 0.115   | 0.209  | 0.107         | 0.176  | 0.064       | 0.087  | 0.083     | 0.120  | 0.083   | 0.121  | 0.074  | 0.099  | 0.082    | 0.131    |
> | GeoTDM 1.9M (all→each)   | 0.091   | 0.164  | 0.024   | 0.040  | 0.104   | 0.191  | 0.097         | 0.164  | 0.061       | 0.092  | 0.074     | 0.114  | 0.073   | 0.112  | 0.070  | 0.102  | 0.074    | 0.122    |
> | LAM-SLIDE 2.5M           | 0.059   | 0.098  | 0.021   | 0.032  | 0.087   | 0.167  | 0.073         | 0.124  | 0.037       | 0.058  | 0.047     | 0.074  | 0.045   | 0.075  | 0.050  | 0.074  | 0.052    | 0.088    |
> | LAM-SLIDE 2.1M           | 0.064   | 0.104  | 0.023   | 0.033  | 0.097   | 0.182  | 0.084         | 0.141  | 0.044       | 0.067  | 0.053     | 0.081  | 0.054   | 0.086  | 0.054  | 0.079  | 0.059    | 0.097    |
> | LAM-SLIDE 1.7M           | 0.074   | 0.117  | 0.025   | 0.037  | 0.110   | 0.195  | 0.097         | 0.159  | 0.053       | 0.074  | 0.063     | 0.091  | 0.064   | 0.094  | 0.064  | 0.089  | 0.069    | 0.107    |

---

> > ### Comment · Reviewer_fy2s · 2025-08-06
> >
> > Thank you for the rebuttal. The author has addressed all my questions.

---

> > > ### Author Response · Authors · 2025-08-07
> > >
> > > Thank you for acknowledging that your concerns have been addressed. We appreciate your suggestions regarding identifier design choices which have helped us to refine and improve our manuscript.

---

### Decision · Program_Chairs · 2025-09-17

**Decision:**

Accept (poster)

**Comment:**

This submission proposes LaM-SLidE, a novel framework for modeling spatial dynamical systems using latent generative modeling with identifier (ID)-based entity traceability. The core contribution is an encoder–latent approximator–decoder architecture that incorporates learnable IDs to ensure entity traceability while leveraging fixed-size latent representations. This allows modern scalable generative models (flow matching, diffusion) to be applied to systems with variable numbers of entities. The approach is validated across multiple domains (molecular dynamics, N-body systems, pedestrian trajectories, basketball plays), showing competitive or superior results compared to baselines such as GeoTDM and specialized GNN methods, while requiring substantially fewer function evaluations.

Overall, this is a technically solid and novel paper that makes a meaningful contribution to generative modeling of dynamical systems. While the ETH-UCY results remain less competitive than specialized trajectory forecasting methods, the strength of this work lies in its generalizable architecture that unifies modeling across diverse domains while significantly improving efficiency. The authors have been highly responsive in addressing concerns, adding ablations, clarifying methodology, and expanding the discussion of limitations and related work.

The main weaknesses (clarity, limited ablations, and missing baseline discussion) have been sufficiently mitigated through the rebuttal and revisions. Concerns about scope (tracking, dynamic entities, permutation invariance) are valid but appropriately acknowledged as future directions rather than fatal flaws.